# Construction of Attenuated Strains for Red-Spotted Grouper Nervous Necrosis Virus (RGNNV) via Reverse Genetic System

**DOI:** 10.3390/v14081737

**Published:** 2022-08-06

**Authors:** Yingying Lei, Yu Xiong, Dagang Tao, Tao Wang, Tianlun Chen, Xufei Du, Gang Cao, Jiagang Tu, Jinxia Dai

**Affiliations:** 1State Key Laboratory of Agricultural Microbiology, Huazhong Agricultural University, Wuhan 430070, China; 2College of Veterinary Medicine, Huazhong Agricultural University, Wuhan 430070, China; 3Institute of Neuroscience, State Key Laboratory of Neuroscience, Center for Excellence in Brain Science and Intelligence Technology, Shanghai Research Center for Brain Science and Brain-Inspired Intelligence, Chinese Academy of Sciences, Shanghai 200031, China; 4University of Chinese Academy of Sciences, Beijing 100049, China; 5College of Fisheries, Huazhong Agricultural University, Wuhan 430070, China

**Keywords:** RGNNV, reverse genetics, B2 protein, synonymous mutation

## Abstract

The nervous necrosis virus (NNV) mainly attacks the central nervous system of fish to cause viral nervous necrosis, which is an acute and serious prevalent disease in fish. Among different genotypes of NNV, red-spotted grouper nervous necrosis virus (RGNNV) is the most widely reported, with the highest number of susceptible species. To better understand the pathogenicity of RGNNV, we first developed a reverse genetic system for recombinant RGNNV rescue using B7GG and striped snakehead (SSN-1) cells. Furthermore, we constructed attenuated RGNNV strains rRGNNV-B2-M1 and rRGNNV-B2-M2 with the loss of B2 protein expression, which grew slower and induced less Mx1 expression than that of wild-type RGNNV. Moreover, rRGNNV-B2-M1 and rRGNNV-B2-M2 were less virulent than the wild-type RGNNV. Our study provides a potential tool for further research on the viral protein function, virulence pathogenesis, and vaccine development of RGNNV, which is also a template for the rescue of other fish viruses.

## 1. Introduction

Viral nervous necrosis (VNN) caused by betanodaviruses of the family *Nodaviridae*, has been listed as an important prevalent fish disease by the Office International des Epizooties due to its extremely high infectivity and enormous economic losses. Nervous necrosis virus (NNV), a member of the *Betanodavirus* genus, mainly infects larvae and juveniles of more than 177 marine and freshwater fish species with high mortality rates exceeding 95% in severe outbreaks [1,2,3]. Red-spotted grouper nervous necrosis virus (RGNNV), the major causative agent of marine fish VNN disease in China and Southeast Asia, was first isolated and detected in the affected tissues in China [4,5,6]. The genome of the RGNNV consists of two positive single-stranded linear RNA segments (RNA1 and RNA2). RNA1 (3.1 kb) encodes RNA-dependent RNA polymerase (RdRp), also named protein A, which is responsible for the replication of virus [7]. RNA2 (1.4 kb) encodes the capsid protein (Cp), which is the sole structural protein of viral particle [8]. A subgenome of RNA1, called RNA3, located at the 3′-terminus of RNA1, encodes two small non-structural proteins B1 (111 aa) and B2 (75 aa). B2 binds to newly synthesized viral double-stranded RNA, therefore preventing host RNA interference-mediated cleavage, and can also promote mitochondria-mediated cell death induced by hydrogen peroxide production [9,10,11,12].

Innate immunity plays a key role against NNV before the subsequent adaptive immunity. Betanodavirus and its host have a complex interaction during virus infection [13]. The host responds to betanodavirus infection through initiating an intracellular innate immune response, which can limit virus replication, increase antiviral gene expression and induce secretion of inflammatory cytokine including interleukins (IL-1β), tumor necrosis factor (TNF), and myxovirus resistance protein (Mx protein) [14]. Endonuclease G (Endo-G), an apoptotic nuclease, was up-regulated after RGNNV infection in SSN-1 cells, and was supposed to be involved in cell apoptosis induced by RGNNV [15]. Positive correlations between C-reactive protein-like protein (CRP) levels and viral infections have been established in fish [16,17]. To date, the pathogenicity of RGNNV remains to be understood. Meanwhile, a commercial vaccine against RGNNV is still desperately needed, and potential vaccine candidates against RGNNV for large-scale application in fishes are still at the laboratory level [18,19,20,21,22].

The reverse genetic system has been widely used in various virology studies, and is a powerful tool to study the pathogenic mechanism and develop vaccines of viruses [23,24,25]. It has been established for several fish RNA viruses such as viral hemorrhagic septicemia virus (VHSV) [26], infectious hematopoietic necrosis virus (IHNV) [27], sleeping disease virus (SDV) [28], as well as snakehead vesiculovirus (SHVV) [29]. Alphanodavirus including infectious flock house virus (FHV), pariacoto virus (PaV), and nodamura virus (NoV) were developed reverse genetics [30,31,32]. Both RNA-based and DNA-based genetic systems are now available for several betanodavirus species including RGNNV [9,33,34,35,36,37]. The recombinant RGNNV harboring site-specific mutations in the capsid protein sequence caused a significant decrease in virulence [35]. Due to the important role of B2 protein [9,10,11,12], constructing RGNNV without B2 expression will be another potential attenuated strain for vaccine development and pathogenic mechanism study of RGNNV. However, the reverse genetic system for recombinant RGNNV lacking B2 protein has not been successfully established.

In the present study, we aimed to construct attenuated RGNNV without B2 expression based on developing a reverse genetic system for the rescue of RGNNV. This study will be helpful for future understanding of molecular mechanism for RGNNV pathogenicity as well as development of genetic engineering vaccine based on the B2 mutant virus.

## 2. Materials and Methods

### 2.1. Cells and Virus

B7GG cells, which originated from BHK-21 cells and express T7 RNA polymerase, rabies virus G and histone-tagged GFP [38], were kind gifts from Drs. Fuqiang Xu (Shenzhen Institutes of Advanced Technology, Chinese Academy of Sciences) and Edward Callaway (The Salk Institute for Biological Studies, La Jolla, CA, USA). B7GG cells were grown in Dulbecco’s Modified Eagle Medium (DMEM) supplemented with 10% fetal bovine serum (FBS, 16140-071, Gibco, Grand Island, NY, USA), 10 U/mL penicillin, and 10 mg/mL streptomycin at 37 °C with 5% CO_2_ or in Opti-MEM™ Reduced Serum Medium (31985-070, Gibco, Grand Island, NY, USA). Striped snakehead fish (SSN-1) cells were grown in Leibovitz’s L-15 medium (Gibco) supplemented with 10% FBS (Gibco), 10 U/mL penicillin, and 10 mg/mL streptomycin at 27 °C. Wild-type RGNNV (wtRGNNV) was isolated from diseased red-spotted grouper and stored in our lab [4]. The wtRGNNV was replicated in SSN-1 cells grown in L-15 medium with 2% FBS at 27 °C, and the supernatant was harvested when at least 80% of infected cells show cytopathic effect (CPE), then clarified and stored at −80 °C for further processing.

### 2.2. Antibodies and Reagents

The antibody against B2 protein of RGNNV was produced by aBIOTECH company (Jinan, China) through immunizing rabbits with B2 protein expressed in *E. coli*. The affinity and specificity of anti-B2 serum were confirmed by positive band in wtRGNNV-infected SSN-1 cells, but no signal was detected in non-infected SSN-1 cells through Western blotting assay (Appendix A). Other antibodies and reagents used in the current study were mouse anti-β-actin (66009-1, ProteinTech Group, Chicago, IL, USA), horseradish peroxidase (HRP) goat anti-mouse IgG (AS003, Abclonal, Wuhan, China), HRP goat anti-rabbit IgG (AS014, Abclonal, Wuhan, China), Alexa Fluor^®^ 488 goat anti-mouse IgG (A-11001, Invitrogen, Grand Island, NY, USA), 4′,6-diamidino-2-phenylindole (DAPI) (C1002, Beyotime, Shanghai, China), proteinase inhibitor cocktail (P8340, Sigma, Darmstadt, Germany), phosphatase inhibitor cocktail 3 (P0044, Sigma, Darmstadt, Germany).

### 2.3. Construction of RNA1 and RNA2 Expression Plasmids

The open reading frames (ORFs) of RNA1 and RNA2 from RGNNV with hepatitis delta virus ribozyme element sequence (RNA1-HDVrz and RNA2-HDVrz) were synthesized (Beijing tsingke Biotech, Beijing, China) and amplified by PCR, then cloned into pcDNA3.1-CMV-eGFP vector between *Bgl* II and *Eco*R I sites to obtain pRGNNV-T7-RNA1-CMV-eGFP and pRGNNV-T7-RNA2-CMV-eGFP, respectively [29]. All insertions used in this study were confirmed by Sanger sequencing (Shanghai Sangon Biotech, Shanghai, China) with the upper primer of T7 promoter (T7p-F: ctcagatcttaatacgactcactatagg) and terminal primer of inserted fragment (RNA1-T7t-R: taccgaattcccgctgagcaataactagc; RNA2-T7t-R: taccgaattcccgctgagcaataac). The sequences for RNA1 and RNA2 can be found in the Appendix A (Appendix A).

### 2.4. In Vitro Site-Directed Mutagenesis

Mutagenesis of start codon from ATG to ACG for B2 protein was performed by PCR and cloned into pRGNNV-T7-RNA1-CMV-eGFP vector to obtain pRGNNV-T7-RNA1-B2-M1-CMV-eGFP. Mutagenesis primers for pRGNNV-T7-RNA1-B2-M1-CMV-eGFP were: B2-M1-F, 5′-caaaactagcaaggacggaacaaatccaacaag-3′; B2-M1-R, 5′-gtccttgctagttttgttttgactaggaccggcttg-3′. Mutagenesis of the upstream start codon (15 amino acids upstream of the start codon of B2 protein) from ATG to ACG was performed by PCR and cloned into pRGNNV-T7-RNA1-B2-M1-CMV-eGFP to obtain pRGNNV-T7-RNA1-B2-M2-CMV-eGFP. Mutagenesis primers for pRGNNV-T7-RNA1-B2-M2-CMV-eGFP were: B2-M2-F, 5′-tgtctgtcgcattagacggggagatccaagccg-3′; B2-M2-R, 5′-gtctaatgcgacagacatctttggttc-3′. The mutant plasmids were sequenced to confirm incorporation of the desired mutation.

### 2.5. Reverse Genetics

6.0 × 10^5^ B7GG cells were seeded in 25 cm^2^ cell culture flask (Costar, Washington, DC, USA) and grown to 80% confluence (9.0 × 10^5^) at 37 °C in 5% CO_2_. Cells were co-transfected with the supporting plasmids pRGNNV-T7-RNA2-CMV-eGFP (25 μg) together with pRGNNV-T7-RNA1-CMV-eGFP (25 μg) or the mutant plasmids (25 μg) (pRGNNV-T7-RNA1-B2-M1-CMV-eGFP or pRGNNV-T7-RNA1-B2-M2-CMV-eGFP) using TransIntro^TM^ EL transfection reagent (FT201-01, TransGen Biotech, Beijing, China) according to the manufacturer’s instructions. At 12 h post-transfection, the medium was replaced by 30 mL of DMEM with 5% FBS and cells were cultured at 27 °C for 3 days. Then cells were freeze-thawed three times and then centrifuged at 10,000× *g* for 5 min. The supernatant was subsequently incubated with fresh SSN-1 cells at 27 °C for three blind passages (3 days/passage) and the cytopathic effect (CPE) was observed daily.

### 2.6. Virus Infection, Titration and Kinetics of Viral Replication Assays

SSN-1 cells were incubated with wtRGNNV, rRGNNV, rRGNNV-B2-M1 or rRGNNV-B2-M2 virus. At 72 h post-infection (hpi), both the supernatants and cells were collected for viral titration by 50% tissue culture infective dose (TCID_50_).

For kinetics of viral replication assay, SSN-1 cells were incubated with wtRGNNV, rRGNNV, rRGNNV-B2-M1 or rRGNNV-B2-M2 virus at a MOI of 5. Then cells were collected at 0, 24, 48, and 72 hpi for viral Cp RNA detection by RT-qPCR with primers listed in Table 1. Three replicates for each treatment were analyzed.

### 2.7. RNA Extraction, RT-PCR and RT-qPCR Assays

Total RNA was isolated using Trizol reagent (15596026, Invitrogen, Grand Island, NY, USA) according to the manufacturer’s instructions. The quality and quantity of total RNA were checked by NanoDrop ND-2000 spectrophotometer (Thermo Scientific). 0.5 µg of total RNA was processed directly to cDNA with HiScript III 1st strand cDNA synthesis kit (R312, Vazyme Biotech, Nanjing, China) following the manufacturer’s instructions. The RT-PCR for RNA1 and RNA2 of RGNNV were performed with primers RdRp-RT and Cp-RT, respectively. RT-qPCR reactions were performed in a 10 µL volume of qPCR SYBR Green Master Mix (A25742, Thermo Fisher Scientific,). All the reactions were triplicated and performed in ABI 7300 system (Applied Biosystems). Amplification was initiated at 95 °C for 5 min, followed by 40 cycles of 95 °C for 15 s, 58 °C for 20 s, and 72 °C for 20 s. To confirm specificity of amplification, the PCR products from each primer pair were subjected to a melting curve analysis and electrophoresis in 2% agarose gel. All primers used for RT-PCR and RT-qPCR are listed in Table 1. 18S or β-actin were used as internal control. The relative RNA levels were normalized to the corresponding controls using the 2^−ΔΔCt^ method [39,40]. Three independent experiments were conducted for statistical analysis.

### 2.8. Western Blotting Assay

5 × 10^5^ of B7GG or SSN-1 cells were seeded into six well plates and grown to 80–90% confluence. 3 μg of recombinant plasmid pRGNNV-T7-RNA1-CMV-eGFP, pRGNNV-T7-RNA1-B2-M1-CMV-eGFP or pRGNNV-T7-RNA1-B2-M2-CMV-eGFP was transfected into B7GG cells using TransIntro^TM^ EL transfection reagent (FT201-01, TransGen Biotech, Beijing, China). SSN-1 cells were infected with wtRGNNV, rRGNNV, rRGNNV-B2-M1 or rRGNNV-B2-M2 at an MOI of 5. B7GG cells at 36 h post-transfection and SSN-1 cells at 3 days post-infection (dpi) were washed with PBS and then lysed with RIPA lysis buffer (50 mM Tris-HCl, pH 7.5, 150 mM NaCl, 0.05% Nonidet P-40) supplemented with 1% protease inhibitor cocktail (P8340, Sigma, Darmstadt, Germany). The whole cell lysates were centrifuged at 10,000× *g* for 10 min at 4 °C to remove debris and boiled with SDS loading buffer for 5–10 min. The proteins were separated by 15% SDS-PAGE gels, and then transferred to a polyvinylidene fluoride membranes (Merck Millipore, Darmstadt, Germany). The membrane was blocked with 5% (m/v) fat-free dry milk in TBST buffer (25 mM Tris-HCl, 150 mM NaCl, and 0.1% Tween-20 (pH 7.5)) for 2 h at room temperature. Rabbit anti-B2 polyclonal antibody (1:500, prepared in our lab) was applied to the membrane overnight at 4 °C. The membrane was washed three times with TBST and then incubated with HRP-conjugated goat anti-rabbit secondary antibody (1:3000, ABclonal, Wuhan, China) for 1 h at room temperature. The specific proteins were visualized using electro-chemiluminescence ECL kit with Image Lab software 4.0.1 (1705060, Bio-Rad, California, USA). For quantification analysis, bands for B2 and β-actin were quantified using the threshold function of Image J. Three independent experiments for B2 Western blotting were analyzed for quantification. Statistical significance was analyzed using the Student’s *t*-test (* *p* < 0.05; ** *p* < 0.01; n.s. = not significant).

### 2.9. Immunofluorescence Assay

5 × 10^4^ of SSN-1 cells were seeded into a 24-well plate. After 24 h, cells were infected with wtRGNNV, rRGNNV, rRGNNV-B2-M1, or rRGNNV-B2-M2 at an MOI of 5 at 27 °C for 12 h and then replaced by media with L-15 medium containing 2% FBS for another 24 h. Then, infected cells were fixed with 4% paraformaldehyde in phosphate-buffered saline (PBS) at room temperature for 15 min, and then perforated with 0.5% Triton X-100 in PBS for 15 min. Cells were blocked using 3% BSA in PBS for 30 min, and then incubated with rabbit anti-B2 polyclonal antibody at 27 °C for 1 h. After three times washing with PBS at room temperature, cells were incubated with FITC-conjugated goat anti-rabbit IgG (1:500, Southern Biotech, Birmingham, AL, USA) at 27 °C for 1 h. Images of immunofluorescence were obtained by Olympus fluorescent microscope (Olympus Corporation, Tokyo, Japan). The percentage of B2 positive cells and mean fluorescent intensity of B2 positive cells were measured via Image J. Student’s t-test was used for comparison of the quantification data from three independent experiments. The *p* value < 0.05 was considered statistically significant (*).

### 2.10. Cytopathic Effect Measurement and Expression Analysis on Endo-G and Mx1 

To measure the cytopathic effect of rRGNNV, rRGNNV-B2-M1, and rRGNNV-B2-M2, we infected SSN-1 cells with the three constructed viruses with non-infected SSN-1 cells as control. For each virus infection, three repeats were obtained. Then we randomly selected three visual fields to capture the images for each culture well under 40X microscope. We counted the cell number of vacuolar cytopathic lesions in each image for final statistical analysis. Student’s t-test was used for comparison of the quantification data from three independent experiments (* *p* < 0.05; ** *p* < 0.01; *** *p* < 0.001; **** *p* < 0.0001).

To analyze the mRNA expression of cytotoxic protein Endo-G and antiviral pathway related protein Mx1 induced by RGNNV infection, SSN-1 cells infected by wtRGNNV or three recombinant RGNNV at 3 dpi were collected to extract total RNA and subjected to RT-qPCR analysis with primers for Endo-G and Mx1 listed in Table 1.

### 2.11. Zebrafish Experiment

Zebrafish were handled in compliance with the local animal welfare regulations and maintained according to standard protocols. Embryos were grown at 25–30 °C in egg water (60 μg/mL Instant Ocean salts). For viral load and cytokine expression analysis, wild-type AB zebrafish (*Danio rerio*) larvae (naturally hatched germ-free) at 4 days post fertilization (4 dpf) were infected by static immersion with wtRGNNV, rRGNNV, rRGNNV-B2-M1 or rRGNNV-B2-M2 at concentration of 1.45 × 10^6^ TCID_50_/mL, 1.45 × 10^5^ TCID_50_/mL, 1.45 × 10^4^ TCID_50_/mL for 30 h. Then total RNA was isolated from zebrafish larvae at 30 hpi and followed by RT-qPCR with primers for Cp, CRP, and TNF-α listed in Table 1. Zebrafish β-actin was used as internal control. Zebrafish larvae without virus infection was set as control. Ten zebrafish larvae for each group and three repeats were performed.

For in vivo cytopathic effect analysis, transgenic zebrafish Tg (elavl3: GCaMP6s) jf4;nacre larvae (naturally hatched germ-free) at 4 dpf were kept under anesthesia in egg water containing 0.02% buffered 3-aminobezoic acid ethyl ester (tricaine; Sigma-Aldrich). Then 4 × 10^3^ TCID_50_/larvae of wtRGNNV, rRGNNV, rRGNNV-B2-M1 or rRGNNV-B2-M2 viruses were injected into midbrain-hindbrain boundary area of larvae through micro-injection as previously described [41]. Followed infection, zebrafish larvae at 8 dpf were cryo-sectioned at coronal plane for Cp immunostaining or directly imaged by confocal microscopy. Zebrafish larvae without virus infection was set as control. Three zebrafish larvae for each group were performed.

### 2.12. Statistics Analysis

Numerical data were analyzed by GraphPad Prism 7.0 software (GraphPad Software, Inc) from three independent experiments shown as mean ± SD. Evaluation of the significance of differences between groups was performed using Student’s t-test, and *p* value < 0.05 was considered statistically significant (*).

## 3. Results

### 3.1. Construction of the Recombinant RGNNV (rRGNNV) Virus

To establish the reverse genetic system of RGNNV, the RNA1 and RNA2 sequence of RGNNV flanked with T7 promoter containing two additional guanine residues at its 5′ end and hepatitis delta virus ribozyme (HDVrz) sequences at its 3′ end was cloned step by step into the expression vector pcDNA3.1-CMV-eGFP (Figure 1 and Appendix A). The constructed plasmids pRGNNV-T7-RNA1-CMV-eGFP and pRGNNV-T7-RNA2-CMV-eGFP were confirmed by correct size of inserted fragments with 3.24 kb and 1.57 kb length, respectively, by *Bgl* II and *Eco*R I double-enzyme digestion (Figure 1A,B). In this study, we rescued recombinant RGNNV (rRGNNV) with viral packaging in B7GG cells based on its high transfection efficiency and then amplification in fish SSN-1 cells based on its high efficiency for RGNNV replication (Figure 1C,D). B7GG cells were co-transfected with the plasmids pRGNNV-T7-RNA1-CMV-eGFP and pRGNNV-T7-RNA2-CMV-eGFP. Then B7GG cells were collected and then freeze-thawed for centrifugation, after which the supernatants were incubated with fish SSN-1 cells for viral replication. For the positive control, SSN-1 cells were infected with wtRGNNV directly. Since RGNNV infection will cause significant cytopathy, we observed the cytopathic effect (CPE) of rRGNNV in SSN-1 cells daily for 3–4 days. As show in Figure 1E, CPE that cells forming vacuolar lesions could be observed in most of SSN-1 cells infected with rRGNNV at 72 hpi. To confirm whether the CPE was caused by the recombinant virus rRGNNV, we measured viral RNA1 and RNA2 using RT-PCR. The RGNNV RNA1 and RNA2 were detected in SSN-1 cells infected with the supernatants of B7GG cells transfected with the two plasmids, but not for the control cells without viral infection, indicating that rRGNNV was successfully rescued in B7GG cells to cause cytopathy (Figure 1F).

### 3.2. Construction of the Recombinant RGNNV-B2-M1 (rRGNNV-B2-M1) and RGNNV-B2-M2 (rRGNNV-B2-M2) Viruses

Of four proteins (RdRp, Cp, B1, B2) expressed by RGNNV, the non-structural protein B2 plays an important role in viral replication and cytopathy [9,10]. Thus, loss of B2 protein will highly probably attenuate the infection of RGNNV. Due to the same reading frame of B2 to RdRp, deletion of B2 sequence will directly prevent RdRp expression, which causes failure of RGNNV packing. Since +1 reading frame of B2 relative to RdRp during translation, synonymous mutation of B2 start codon will only interfere with B2 translation but not RdRp expression. Thus, to obtain an attenuated strain of RGNNV, we tried to construct synonymous mutation of rRGNNV to interfere with B2 but not RdRp expression using the established reverse genetic system (Figure 2A). To this aim, the start codon of the B2 protein was mutated from ATG to ACG and cloned into pRGNNV plasmid to obtain pRGNNV-T7-RNA1-B2-M1-CMV-eGFP (Figure 2A,B). Since there is another ATG at 15 amino acids upstream of the start codon of B2 protein, B2 protein might still be translated. To completely prevent the translation of the B2 protein, further point mutation was performed on the upstream ATG site to obtain the plasmid pRGNNV-T7-RNA1-B2-M2-CMV-eGFP (Figure 2B), which contained double synonymous mutations. To verify the effect of mutation on B2 expression, the plasmid pRGNNV-T7-RNA1-CMV-eGFP, pRGNNV-T7-RNA1-B2-M1-CMV-eGFP or pRGNNV-T7-RNA1-B2-M2-CMV-eGFP was transfected into B7GG cells, and the expression of B2 was detected by Western blotting. For the positive control, SSN-1 cells were infected with wtRGNNV directly. Blank cells were used as negative control. As shown in Figure 2C,D, B2 protein could be detected in SSN-1 cells infected with wtRGNNV or B7GG cells transfected with the plasmid pRGNNV-T7-RNA1-CMV-eGFP. On the contrary, the expression of B2 protein was almost undetectable in cells transfected with mutant plasmid pRGNNV-T7-RNA1-B2-M1-CMV-eGFP or pRGNNV-T7-RNA1-B2-M2-CMV-eGFP, indicating that synonymous mutations of B2 start codon could successfully terminate B2 expression.

Using the same method for rescuing rRGNNV, we tried to rescue rRGNNV-B2-M1 and rRGNNV-B2-M2. The virus titer of wtRGNNV, rRGNNV, rRGNNV-B2-M1, and rRGNNV-B2-M2 was detected by TCID_50_ (Figure 2E). The viral replication kinetics of rRGNNV, rRGNNV-B2-M1, and rRGNNV-B2-M2 was subsequently analyzed on SSN-1 cells and compared to wtRGNNV. SSN-1 cells were infected with wtRGNNV, rRGNNV, rRGNNV-B2-M1 or rRGNNV-B2-M2 at an MOI of 5, then the cells and supernatants were collected at 0, 24, 48 and 72 hpi for viral detection via RT-qPCR. The results showed that rRGNNV, rRGNNV-B2-M1, and rRGNNV-B2-M2 grew slower than wtRGNNV at all stage (*p* < 0.001 for rRGNNV vs. wtRGNNV, rRGNNV-B2-M1 vs. wtRGNNV, and rRGNNV-B2-M2 vs. wtRGNNV at 24, 48, and 72 hpi), but close to wtRGNNV after 48 hpi (Figure 2F). The copy number of rRGNNV-B2-M1 at 48 hpi and rRGNNV-B2-M2 at 24 hpi is significantly lower than that of rRGNNV, but reached to the level of rRGNNV at 72 hpi (Figure 2F). So rRGNNV-B2-M1 and rRGNNV-B2-M2 were successfully rescued.

### 3.3. rRGNNV-B2-M1 and rRGNNV-B2-M2 Viruses Were Attenuated to SSN-1 Cells Than Wild-Type RGNNV

To further evaluate whether the two mutated strains of RGNNV (rRGNNV-B2-M1 and rRGNNV-B2-M2) were attenuated in cells, the CPE induced by the two mutant viruses was compared with that by wtRGNNV and rRGNNV. SSN-1 cells were infected with wtRGNNV, rRGNNV, rRGNNV-B2-M1 or rRGNNV-B2-M2 at an MOI of 5. The CPE of cells infected with rRGNNV-B2-M1 or rRGNNV-B2-M2 was significantly weaker than that of wtRGNNV and rRGNNV-infected cells at 3 dpi (Figure 3A,B). The expression of RGNNV RNA1 and RNA2 were detected in SSN-1 cells infected with rRGNNV-B2-M1, rRGNNV-B2-M2, and wtRGNNV by RT-PCR (Figure 3C). When SSN-1 cells infected with rRGNNV-B2-M1 or rRGNNV-B2-M2, B2 protein was significantly reduced as compared with SSN-1 cells infected with wtRGNNV or rRGNNV as detected by immunofluorescent staining (Figure 3D–F) and Western blotting (Figure 3G). These data demonstrated that the two mutant strains of rRGNNV impaired B2 expression. At the same time, SSN-1 cells were collected to extract total RNA, and the mRNA expression of cytotoxic protein Endo-G and antiviral pathway related protein Mx1 were subjected to RT-qPCR analysis. As shown in Figure 3H,I, the mRNA expression of Endo-G and Mx1 was significantly lower in rRGNNV-B2-M1 and rRGNNV-B2-M2 than in rRGNNV-infected cells. These data demonstrated that two mutant strains were attenuated in SSN-1 cells and induced less cytotoxic effect than that of wild-type RGNNV.

### 3.4. rRGNNV-B2-M1 and rRGNNV-B2-M2 Viruses Were Attenuated to Zebrafish Than Wild-Type RGNNV

Since the rescued viruses rRGNNV-B2-M1 and rRGNNV-B2-M2 caused less CPE and antiviral gene expression in vitro as compared with wtRGNNV and rRGNNV, we further analyzed the in vivo virulence of the two mutant RGNNV strains. The zebrafish larvae (4 dpf) were infected by immersion with wtRGNNV, rRGNNV, rRGNNV-B2-M1, or rRGNNV-B2-M2 at concentrations of 1.45 × 10^6^ TCID_50_/mL, 1.45 × 10^5^ TCID_50_/mL, and 1.45 × 10^4^ TCID_50_/mL for 30 h. No obviously abnormal swimming behavior and mortality were observed in zebrafish larvae by immersion infection for 30 h. The zebrafish viral load and cytokine expression were tested. As shown in Figure 4A, all viral loads decreased along with the drop of titer of the infected viruses, but there was no significant difference of the virus load among mutant, rRGNNV and wtRGNNV for each dilution. Since the immune system of zebrafish larvae was immature and mainly relied on natural immunity, we tested the level of two important inflammation factors, C-reactive protein (CRP) and TNF-α by RT-qPCR. As shown in Figure 4B,C, the wtRGNNV infection increased mRNA level of CRP in zebrafish larvae as compared with that in non-infection control, whereas rRGNNV-B2-M1 and rRGNNV-B2-M2 infection caused a significant decrease in the mRNA expression of CRP as compared to wtRGNNV. In addition, the mRNA expression of CRP also decreased as the virus titer dropped. Similar to CRP, the mRNA level of TNF-α in rRGNNV-B2-M1 and rRGNNV-B2-M2 infected zebrafish was also significantly lower than that in wtRGNNV and rRGNNV-infected zebrafish with higher virus titer (1.45 × 10^6^ TCID_50_/mL and 1.45 × 10^5^ TCID_50_/mL) infection. Based on the decreased level of inflammation factors CRP and TNF-α, it is conceivable that the immune response of mutant strains of RGNNV was weakened.

Finally, we observed the cytopathic effect of the two mutant strains in zebrafish larvae. The mutant strains rRGNNV-B2-M1 and rRGNNV-B2-M2 was injected into the midbrain-hindbrain boundary (MHB) area of transgenic zebrafish Tg (elavl3: GCaMP6s) jf4;nacre in which the promoter of neurodevelopment marker gene elavl3 (HuC) drives the expression of GCaMP6s [42]. No obviously abnormal swimming behavior and mortality were observed in zebrafish larvae at 4 dpi. Viral infection was subsequently detected by anti-capsid immunostaining in coronal sections, and neurotoxicity was monitored by confocal imaging (Figure 4D). All viral injections resulted in retina and brain infections in zebrafish larvae. However, only wtRGNNV and rRGNNV infections could lead to severe central nervous system damage, showing obvious necrosis and vacuolization in the brain, especially in the optic tectum (Figure 4E). As compared with wtRGNNV and rRGNNV, mutant strain rRGNNV-B2-M1 or rRGNNV-B2-M2 did not cause obvious vacuolization and necrosis in brain, indicating a lower cytopathic effect (vacuolating necrosis) of mutant strains at the in vivo level. These results indicated that the mutant strains rRGNNV-B2-M1 and rRGNNV-B2-M2 significantly attenuated its virulence to the zebrafish.

## 4. Discussion

Red-spotted grouper nervous necrosis virus (RGNNV) is a widely spread pathogen in fish that mainly destroys the central nervous system (CNS) and causes viral nervous necrosis (VNN). Reverse genetic system has been widely established for several nodaviruses that function to provide suitable host cells with nodavirus RNA1 and RNA2 [34]. In this report, an efficient genetic system for RGNNV recovery was described. The approach we took toward RGNNV reverse genetics was the creation of DNA-based nodavirus replicon vectors pRGNNV-T7-RNA1-CMV-eGFP and pRGNNV-T7-RNA2-CMV-eGFP that could be directly introduced into T7 RNA polymerase-expressing B7GG cells to liberate infectious nodavirus. Two additional 5′ GG residues following T7 promoter were added to improve transcription initiation by T7 RNA polymerase [32,33,43]. Thus, we tried to rescue rRGNNV using B7GG cells based on its high transfection efficiency and fish SSN-1 cells based on its high efficiency for RGNNV replication. Compared with previously established NNV reverse genetics [34], our plasmids contain EGFP gene which was used to determine whether the virus was successfully rescued or not.

Many laboratories worldwide have been mainly devoted to the development of new vaccine approaches based on inactivated, subunit, and live attenuated vaccines to prevent viral diseases in aquaculture [44,45,46]. A previous study has tried start codon mutation of B2 for greasy grouper nervous necrosis virus (GGNNV), but failed to obtain an infectious GGNNV B2 mutant due to none virus detection for the first passage and reverse mutation of B2 start codon after five consecutive supernatant passages [9]. Our current work first obtained infectious rRGNNV-B2-M1 and rRGNNV-B2-M2 mutants using reverse genetic system. However, both viral titer and replication kinetics of rRGNNV-B2-M1 and rRGNNV-B2-M2 were lower than that of wtRGNNV, and at some timepoints lower than that of rRGNNV although they reached equal level at 72 hpi, indicating that the viral replication kinetics of rRGNNV-B2-M1 and rRGNNV-B2-M2 were probably weakened. Since NNV B2 prevent host RNA interference-mediated cleavage by binding to newly synthesized viral double-stranded RNA [9], B2 mutant probably affected the expression of viral RdRp protein, thus impaired the virus growth. However, RNA2 copy number of all three recombinant RGNNV (rRGNNV, rRGNNV-B2-M1 and rRGNNV-B2-M2) at 72 dpi reached equal (Figure 2F), so B2 mutant should not interfere with rRGNNV growth much, but only affect virus growth a little at the early stage. It is noteworthy that rRGNNV grows slower than wtRGNNV although they are genetically identical (Figure 2F); the exact reason for this is unknown. We sequenced rRGNNV to confirm that no mutation was found in rRGNNV. It might be due to epigenetic modification difference of viral RNA for recombinant RGNNV as compared to wtRGNNV, since epigenetic modification can affect virus including RNA virus by modulating the replication and antiviral immune responses [47,48,49,50]. Meanwhile, two mutation strains attenuated the cytotoxic effect and antiviral genes expression in vitro and virulence to the zebrafish in vivo. Lower cytotoxic effect of mutant strains (rRGNNV-B2-M1 and rRGNNV-B2-M2) in SSN-1 cells and zebrafish is consistent with B2 function, since NNV B2 protein induces mitochondria-mediated necrotic cell death [10,11,12]. Taken together, our recombinant mutant RGNNVs (rRGNNV-B2-M1 and rRGNNV-B2-M2) are greatly attenuated as compared with wtRGNNV. A live attenuated vaccine based on B2 mutant against RGNNV could be more worthy of consideration in terms of cost, protective efficacy, and ease of administration (bath immersion instead of injection for traditional vaccines). However, it is necessary to test the virulence and immune protection on grouper as an attenuated vaccine.

Due to lower virulence of our attenuated mutant strains (rRGNNV-B2-M1 and rRGNNV-B2-M2), it will be promising to develop effective viral tracer based on these attenuated rRGNNV for in vivo neural labeling in fishes, especially zebrafish, which is a fascinating vertebrate animal model for neuroscience research but still lacking effective viral tracer. Based on the principle of split-eGFP [51], we tried incorporating C-terminal eGFP (37 amino acid peptides) sequence to 3′-end of the RNA1 at the pRGNNV-T7-RNA1-CMV-eGFP plasmid, and combined it with reporter zebrafish with complemented eGFP (N-terminal eGFP) expression to realize neural labeling. However, we failed to obtain the infectious virus (data not shown), which might be due to the failure of RNA1 expression interfered by insertion of the partial eGFP sequence. Shorter tag insertion such as HA tag after RNA1 may be considerable to avoid the possible failure of virus saving in future development of NNV-based viral tracers.

## 5. Conclusions

Our study constructed a highly efficient reverse genetic system to generate RGNNV using B7GG and fish SSN-1 cells. We also reported the successful rescue of attenuated viruses rRGNNV-B2-M1 and rRGNNV-B2-M2. This reverse genetic system will provide a powerful tool for the analysis of RGNNV dissemination and pathogenesis, as well as preparation for genotype-matched RGNNV attenuated vaccines. Further studies are needed to generate a more suitable recombinant virus rRGNNV for screening of antiviral drugs. It is also promising to develop a neural tracer based on attenuated rRGNNV for neuroscience study.

## Figures and Tables

**Figure 1 viruses-14-01737-f001:**
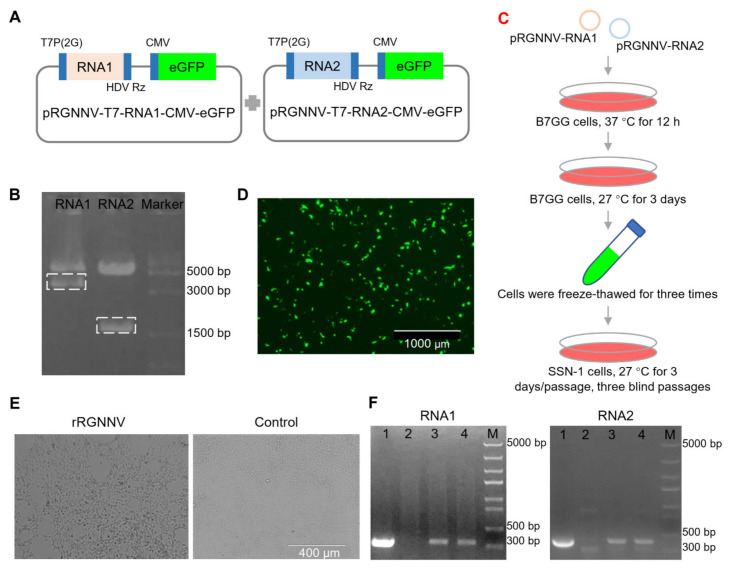
Construction and generation of rRGNNV. (**A**) Schematic representation of the construction for pRGNNV-T7-RNA1-CMV-eGFP and pRGNNV-T7-RNA2-CMV-eGFP encoding the full-length RNA1 and RNA2 of RGNNV, respectively. (**B**) Confirmation of the constructed plasmid pRGNNV-T7-RNA1-CMV-eGFP and pRGNNV-T7-RNA2-CMV-eGFP by restriction digestion with enzyme *Bgl* II and *Eco*R I. The predicted products were about 3100 and 1400 bp, respectively. (**C**) Flow chart of rRGNNV virus rescue. (**D**) Rescuing rRGNNV in B7GG cells. B7GG cells were transfected with plasmids pRGNNV-T7-RNA1-CMV-eGFP and pRGNNV-T7-RNA2-CMV-eGFP, then cultured for 3 days. (**E**) Replication of rRGNNV and its cytopathic effect in SSN-1 cells. The supernatants collected from (**D**) were incubated with fresh SSN-1 cells and the CPE was observed daily. (**F**) Confirmation of rRGNNV infection in SSN-1 cells by RNA1 and RNA2 RT-PCR. RGNNV RNA1 (**left**) and RNA2 (**right**) were detected by RT-PCR for the cells collected from (**E**), lane 1: SSN-1 cells with wtRGNNV infection were used as positive control; lane 2: sample from the right panel of (**E**) was used as negative control; lane 3 and 4: sample form the left panel of (**E**); M: DNA marker.

**Figure 2 viruses-14-01737-f002:**
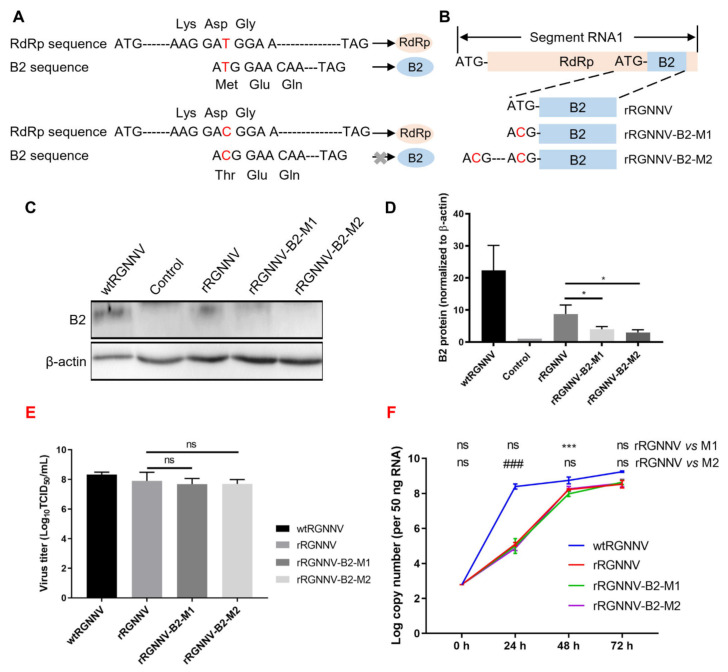
Construction of rRGNNV-B2-M1 and rRGNNV-B2-M2 viruses. (**A**,**B**) Schematic representation of the construction of pRGNNV-T7-RNA1-B2-M1-CMV-eGFP and pRGNNV-T7-RNA1-B2-M2-CMV-eGFP. (**C**,**D**) B2 expression detected by Western blotting. B7GG cells were transfected with plasmid pRGNNV-T7-RNA1-CMV-eGFP, pRGNNV-T7-RNA1-B2-M1-CMV-eGFP or pRGNNV-T7-RNA1-B2-M2-CMV-eGFP for 36 h. SSN-1 cells infected with wtRGNNV directly were used as positive control. Blank cells were used as negative control. B2 was detected by Western blotting, β-actin was used as the internal control. (**E**) The virus titer of wtRGNNV, rRGNNV, rRGNNV-B2-M1, and rRGNNV-B2-M2 was measured using TCID_50_. Both supernatants and SSN-1 cells were collected at 72 hpi for viral titration. (**F**) Viral replication kinetics of recombinant RGNNV. SSN-1 cells were infected with 5 MOI of wtRGNNV, rRGNNV, rRGNNV-B2-M1, and rRGNNV-B2-M2, respectively. Cells were collected at 0, 24, 48 and 72 hpi for viral Cp RNA detection by RT-qPCR. Data shown in (**D**–**F**) are mean ± SD of three independent experiments. * *p* < 0.05; *** *p* < 0.001; ^###^
*p* < 0.001; ns, not significant; both by Student t-test. All experiments were performed in triplicate.

**Figure 3 viruses-14-01737-f003:**
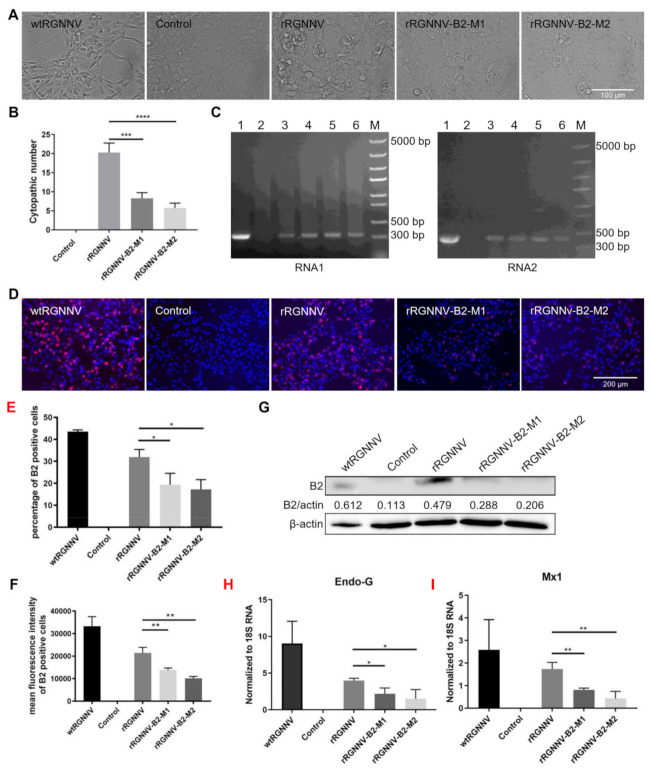
Biological characteristics of wtRGNNV, rRGNNV, rRGNNV-B2-M1, and rRGNNV-B2-M2 viruses. (**A**) CPE of recombinant RGNNV in SSN-1 cells. SSN-1 cells were incubated with wtRGNNV, rRGNNV, rRGNNV-B2-M1, and rRGNNV-B2-M2, and the CPE was observed at 3 dpi. (**B**) The CPE number statistics (*** *p <* 0.001; **** *p <* 0.0001). (**C**) Detection of RNA1 and RNA2. RGNNV RNA1 (**left**) and RNA2 (**right**) were detected by RT-PCR for the SSN-1 cells from (**A**), lane 1: SSN-1 cells infected with wtRGNNV was used as positive control; lane 2: Blank cells were used as negative control; lane 3 and 4: SSN-1 cells infected with rRGNNV-B2-M1; lane 5 and 6: SSN-1 cells infected with rRGNNV-B2-M2; M: DNA marker. (**D**) Immunostaining of B2 in RGNNV infected SSN-1 cells. SSN-1 cells were infected with wtRGNNV, rRGNNV, rRGNNV-B2-M1, and rRGNNV-B2-M2 at an MOI of 5. After 36 h, B2 was detected by immunofluorescence assay. Blank cells were used as negative control. (**E**,**F**) Statistics of B2 immunofluorescence. * *p <* 0.05, ** *p <* 0.01. (**G**) B2 was detected by Western blotting, and β-actin was used as the internal control. (**H,I**) the mRNA expression of Endo-G and Mx1 were detected by RT-qPCR. * *p <* 0.05; ** *p <* 0.01. Scale bar: 100 µm for (A), 200 µm for (D).

**Figure 4 viruses-14-01737-f004:**
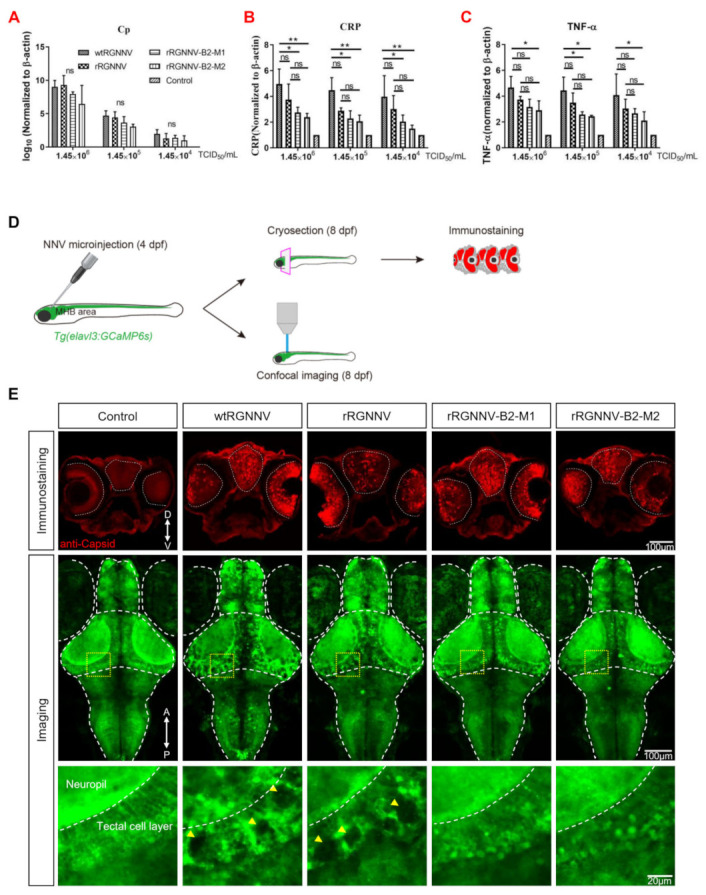
Virulence of wtRGNNV, rRGNNV, rRGNNV-B2-M1 and rRGNNV-B2-M2 viruses in zebrafish. (**A**) Virus load of zebrafish infected with wtRGNNV, rRGNNV, rRGNNV-B2-M1, or rRGNNV-B2-M2 was measured by RT-qPCR of RGNNV. Cp, capsid protein; ns, not significant. (**B**,**C**) The mRNA expression of C-reactive protein (CRP) and TNF-α in zebrafish infected with wtRGNNV, rRGNNV, rRGNNV-B2-M1, or rRGNNV-B2-M2 by RT-qPCR. * *p <* 0.05; ** *p <* 0.01; ns, not significant. (**D**) Schematics of the experimental procedure for viral infection and viral detection. (**E**) Immunofluorescence images of RGNNV capsid protein (Cp) (**top**) and projected transgenic fluorescence-image stacks (**middle**) from larval zebrafish without viral injection or with wtRGNNV, rRGNNV, rRGNNV-B2-M1 or rRGNNV-B2-M2 virus injection. Scale bars, 100 µm. D, dorsal; V, ventral. Enlarged view of the boxed regions in optic tectum were shown at the (**bottom**). Yellow arrowheads indicate massive tissue vacuolization. Scale bar, 20 µm. A, anterior; P, posterior. Data shown in (**A**–**C**) are mean ± SD of three independent experiments. * *p* < 0.05; ** *p* < 0.01; ns, not significant; by two-way ANOVA. All experiments were performed in triplicate.

**Table 1 viruses-14-01737-t001:** Primers for RT-PCR and RT-qPCR.

Primer	Sequence (5′-3′)
RdRp-RT-F	cagccaagtactgtgtccggagag
RdRp-RT-R	caggtttgaacggcaagttgc
Cp-RT-F	cgtgtcagtgctgtgtcgct
Cp-RT-R	cgagtcaaccctggtgcaga
qPCR-Mx1-F	gttcatcacaagacaagaaaccatc
qPCR-Mx1-R	cacctcctgtgccatcttca
qPCR-EndoG-F	gcttcccgtctctgtctcac
qPCR-EndoG-R	cctccttaaagtcgcacagc
qPCR-18S-F	gacggacgaaagcgaaagcatt
qPCR-18S-R	agttggcatcgtttatggtcgg
qPCR-Cp-F	tgacgcacctgtgtctaagg
qPCR-Cp-R	acagcgtatcgctggaagat
qPCR-CRP-F	tcgatagggaggtcatcctg
qPCR-CRP-R	gacgcacaggtgagtctgaa
qPCR-TNF-α-F	gcgcttttctgaatcctacg
qPCR-TNF-α-R	tgcccagtctgtctccttct
qPCR-actin-F	atggatgaggaaatcgctg
qPCR-actin-R	atgccaaccatcactccctg

## Data Availability

Not applicable.

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
