# Peer review of "Construction of Attenuated Strains for Red-Spotted Grouper Nervous Necrosis Virus (RGNNV) via Reverse Genetic System"

_viruses, 2022, doi:10.3390/v14081737_

Round 1
Reviewer 1 Report
Dear Editor,
The manuscript entitled “Construction of attenuated strains for red-spotted grouper nervous necrosis virus (RGNNV) via reverse genetic system” by Yingying Lei et al. presents the development of a reverse genetic system for recombinant RGNNV rescue by using B7GG and striped snakehead (SSN-1) cell lines. The authors constructed attenuated RGNNV strains (rRGNNV-B2-M1, rRGNNV-B2-M2) with loss of B2 protein expression, which grew slower, were less virulent and induced less Mx1 expression than that of wild-type RGNNV.
Τhe manuscripts’ objects are quite interesting, the manuscript is well-written and could be accepted for publication after minor revisions. My detailed comments for the authors to consider are provided below:
1. Page 2, line 82: Please define the “extensive cytopathic effect”. To what extended CPE should be to harvest the cell e.g. >70%, >90%?
2. Page 2, lines 85-86: Please describe the method for B2 antibody production or add the appropriate reference.
3. Page 3, line 127: Please add the reference.
4. Page 4, line 141: Please provide b-actin primers sequence.
5. Page 6, line 215: Figure 1D does not contain images for both plasmids. Please either add an image for both or correct the text by associating the figure with the correct plasmid.
6. Page 6, lines 233-235: Please rephrase this sentence is a little bit confusing.
7. Figure 2E, 2F: Please add SD in figures.
8. Figure 3B: Please explain how cytopathic number is calculated.
9. Page 11, line 374: references 36 and 37 are old. Please are more recent references on fish vaccines, you will find plenty written the last 3 years.
Author Response
The manuscript entitled “Construction of attenuated strains for red-spotted grouper nervous necrosis virus (RGNNV) via reverse genetic system” by Yingying Lei et al. presents the development of a reverse genetic system for recombinant RGNNV rescue by using B7GG and striped snakehead (SSN-1) cell lines. The authors constructed attenuated RGNNV strains (rRGNNV-B2-M1, rRGNNV-B2-M2) with loss of B2 protein expression, which grew slower, were less virulent and induced less Mx1 expression than that of wild-type RGNNV.
Τhe manuscripts’ objects are quite interesting, the manuscript is well-written and could be accepted for publication after minor revisions. My detailed comments for the authors to consider are provided below:
Thank you very much for your comments and suggestion. We have made revision of our manuscript according to your suggestion and responded to all the comments in this letter. All changes have been tracked in our revised manuscript. Please see our point-by-point response in the following.
- Page 2, line 82: Please define the “extensive cytopathic effect”. To what extended CPE should be to harvest the cell e.g. >70%, >90%?
As suggested, we have defined the extent of CPE to harvest cells (“the supernatant was harvested when at least 80% infected cells show cytopathic effect (CPE)” in our revised manuscript, line 95-96).
- Page 2, lines 85-86: Please describe the method for B2 antibody production or add the appropriate reference.
We asked the aBIOTECH company (Jinan, China) to produce the B2 antibody. They expressed the full length B2 protein in E. coli and use it to immunized rabbit, finally obtained the anti-B2 polyclonal antibodies. The affinity and specificity of B2 antibody were tested by western blotting assay as shown in the following figure. The band for B2 was detected by anti-B2 antibody in wild-type RGNNV (wtRGNNV) infected SSN-1 cells but not in non-infected (Mock) SSN-1 cells. We have added the description on the B2 antibody production and confirmation in the method part (line 99-103) and the following figure as new supplemental Figure 1 in our revised manuscript.
- Page 3, line 127: Please add the reference.
As suggested by another reviewer, we directly used TCID50/mL for RGNNV titer since nodavirus does not form plaques on cell monolayers in our revised manuscript. So, we didn’t add the reference.
- Page 4, line 141: Please provide b-actin primers sequence.
As suggested, we have provided β-actin primers sequence in revised table 1.
- Page 6, line 215: Figure 1D does not contain images for both plasmids. Please either add an image for both or correct the text by associating the figure with the correct plasmid.
We have corrected the text by associating new Figure S3 with the correct plasmid in our revised manuscript.
- Page 6, lines 233-235: Please rephrase this sentence is a little bit confusing.
To avoid the confusion, we have rewritten the sentence as “While since +1 reading frame of B2 relative to RdRp during translation, synonymous mutation of B2 start codon will only interfere with B2 translation but not RdRp expression.” in our revised manuscript (line 282-284).
- Figure 2E, 2F: Please add SD in figures.
In our revised manuscript, we replaced PFU with TCID50/mL for virus titer of wtRGNNV, rRGNNV, rRGNNV-B2-M1 and rRGNNV-B2-M2. So, we revised Figure 2E with SD.
Since the error bars were shorter than the size of symbols in Figure 2F, it is hard to recognized error bars. In our revised Figure 2F, we removed all symbols so that the error bars can be recognized now.
- Figure 3B: Please explain how cytopathic number is calculated.
To measure the cytopathic effect of rRGNNV, rRGNNV-B2-M1 and rRGNNV-B2-M2, we infected SSN-1 cells with the three constructed viruses, and non-infected SSN-1 cells was set as control. For each virus infection, three repeats were obtained. Then we randomly selected three visual field to capture the images for each culture well under 40X microscope. We counted the cell number of vacuolar cytopathic lesions in each image for final statistical analysis.
- Page 11, line 374: references 36 and 37 are old. Please are more recent references on fish vaccines, you will find plenty written the last 3 years.
As suggested, we have replaced reference 36 and 37 with three more recent references on fish vaccines (listed below) in our revised manuscript.
- Ma J, Bruce TJ, Jones EM, Cain KD. A review of fish vaccine development strategies: conventional methods and modern biotechnological approaches. Microorganisms. 2019;7(11):569.
- Cárdenas C, Guzmán F, Carmona M, Muñoz C, Nilo L, Labra A, et al. Synthetic peptides as a promising alternative to control viral infections in Atlantic salmon. Pathogens. 2020;9(8):600.
- Zeng R, Pan W, Lin Y, He J, Luo Z, Li Z, Weng S, He J, Guo C. Development of a gene-deleted live attenuated candidate vaccine against fish virus (ISKNV) with low pathogenicity and high protection. iScience. 2021;24(7):102750. doi: 10.1016/j.isci.2021.102750.

Reviewer 2 Report
The manuscript entitled “Construction of attenuated strains for red-spotted grouper nervous necrosis virus (RGNNV) via reverse genetic system”, describes the generation of two recombinant RGNNV viruses with point mutations within the B2 sequence. Authors have stablished the reverse genetics protocol to generate those viruses and have tested their infectivity both in vitro and in vivo. The manuscript is well written, although some minor mistakes regarding the use of the English language and important mistakes regarding organization have been detected. I suggest to read carefully the MS to correct English mistakes and to consider the following comments:
General comment
I suggest not to use the term “mRNA” or “transcription” when you refer to viral RNA quantification, RGNNV is a positive RNA virus, and, therefore, distinction between genomic RNA and mRNA (or replication and transcription) is not really possible.
Specific comments
Introduction
Line 45: this information is incomplete, since RNA2 also encodes B1 protein. B1 is also missed in line 229.
Line 48: it is written that “commercially available vaccine against RGNNV is currently lacking”. To my knowledge, there are at least two commercial formalin-inactivated vaccines against RGNNV: ALPHA JECT micro® 1Noda (Paharmaq) and ICTHIOVAC® VNN (Hipra), both for sea bass vaccination.
Line 59: authors claim that the reverse genetic system for RGNNV has not being established. This information is not accurate, recombinant RGNNV viruses have been obtained by reverse genetics by Moreno et al. (2019), and recombinant RGNNV/SJNNV reassortant viruses have also being retrieved by Souto et al. (2015, 2018). As a general comment, a more rigorous reference searching is required in the whole MS.
In this work the transcription of some immunogenes has been analysed in vitro and in vivo; however, information about the relevance of those genes in controlling NNV infection does not appear in the “Introduction” section. This information should be added. The transcription analysis of those genes should be put in contest.
Materials and methods
This section is specially confusing, and important information is missing.
“Antibodies and reagents” section (line 85): if anti-B2 antibodies have been already published, authors should add that reference; if not, details about B2 expression, rabbit immunization procedure, and antibody purification should be added.
“Construction of RNA1 and RNA2 expression plasmids” (line 92): I miss information about the sequencing strategy and primers used for RNA1 and RNA2 sequencing, as well as about the cloning procedure. It may be convenient to add this information as supplementary material.
“Virus infection and titration” (line 121):
Please include information about the MOI used for cell inoculation, and the specific sampling times. This information is in “Results” and should appear in “Material and Methods” section.
This section is named “virus infection and titration”; however, information about RT-qPCR, and western-blot is included. I suggest to either move that information to another section or to change the heading of the section. Nevertheless, information about the sampling time for RT-qPCR and blot analysis should be added.
Line 124: it is said that viral titration has been performed at different times poi, but these results are not in the MS.
I do not see the point of expressing nodavirus titre as PFU. I think it would be more adequate to express it as TCID50/ml, especially considering that nodavirus does not form plaques on cell monolayers. The relationship between PFU and TCID50 in line 127 is widely used for adenovirus, but it may not be correct to use it for nodavirus. Finally, using TCID50 would allow to compare your results with previous publications, which express nodavirus titres as TCID50.
Table 1: primers for endogenous gene amplification should be added.
Table 1 heading: it is written “primers for RT-PCR and RT-qPCR”; however, there is only one pair of primers for the amplification of RNA2 genomic segment. Were those primers used for both, PCR and qPCR, or there is a pair of primers missed in this table? Furthermore, there are two pair of primers for RT reaction (CP and RdRp), whereas only the pair of primers for CP amplification is included.
“RNA extraction, virus RT-PCR and RT-qPCR assays” (line 129):
Line 130: it is said that RNA is extracted at 3 days poi and cDNA was amplified with primers in Table 1 (for viral genome amplification, line 141); however, I have not found results of viral qPCR from samples taken at that time. The only qPCR results are those displayed in Figure 2F (samples collected at 0, 24, 48 and 72 h).
Line 141: it is written that the internal control of amplification is “grouper beta-actin”. This may be a mistake, since SSN-1 cells do not derive from that fish species.
“Western blotting assay” (line 145):
According to line 146, western blotting was conducted on B7GG and SSN-1 cells; however, this information does not agree with that in “virus infection and titration” section, where it is said that western blotting is perform on SSN-1 cells, and B7GG cells are not mentioned.
According to lines 147-148, SNN-1 cells were transfected; this information does not agree with that in line 247, where it is said: “for positive control, SNN-1 cells were infected with RGNNV directly”.
At what time post-transfection or post-infection were the cells collected for western blotting?
“Zebrafish experiment” (line 178):
In line 181 it is said “during viral injection, embryos……” and in line 183 “virus infection of zebrafish larvae….”. did you challenge embryos in addition to larvae?
Add information about the challenge: viral dose (which is in “Results” section), number of larvae, number of replicas, times of sampling, strain of zebrafish…..
Nothing is said in this section about the challenge conducted with the transgenic zebrafish strain or about the immunogene analyses, add this information in this section.
Results
Figure 2D: how was the B2 protein calculated in this graph? I have not found this information in the MS.
Figure 2E: in line 256 and figure legend is said “virus titre by TCID50”. More explanation is required: what has been this titre obtained from? are they samples obtained at a specific time post-infection? Why there is no SD in this graph?
Figure 2F: samples were taken at different times, and in line 130 it is said at 3 days poi.
Standard deviation should appear in this graph. Experimental replicas are required for statistical analyses.
Line 262: it is written that mutant recombinant viruses grow “almost equally” as wtRGNNV. In order to stablish if there are differences in viral multiplication, a statistical test should be used.
I do not understand the difference between the copy number of wtRGNNV and rRGNNV at 24 h p.i. Both viruses are supposed to be genetically identical, and, therefore, they should grow with the same kinetics in SNN-1 cells. Please, explain this result.
Figure 3A: did the authors need a blind passage with any of the recombinant viruses?
Figure 3B: please, explain what is the “cytopathic number” and how it has been calculated
Why the wtRGNNV group is included in Figure 3A and disappears in Figure 3B?
Figure 3C: primers used to amplify RNA2 are missed in Table 1.
Based on this picture, authors claim that the expression of RNA1 and RNA2 was not affected by mutations (line 295). Although this seems to be correct, in my opinion this study should have been conducted by qPCR or, alternatively, the analysis of the intensity of the bands should have been performed. If that analysis has been conducted, this information should appear in “Material and Methods”.
Why have you chosen conventional PCR instead of q-PCR in this assay?
Were those samples collected at 3 dpi? If so, there is a contradiction between the results of RNA2 transcription shown in Figure 3C and Figure 2F. According to Figure 2F, at 3 d p.i. the copy number of RNA2 is similar for all viruses, including the wild-type, whereas the band for the wild type virus is clearly more intense in Figure 3C. Can you explain this?
Figure 3D: in line 283 it is said that immunofluorescence has been performed at 48 h pi, is that correct? if so, please correct the information in “Material and Method”, where it is said that this technique was applied 24 h pi (line 167).
I think that the expression of B2 should be the same in cells inoculated with wtRGNNV and rRGNNV, why is lower for rRGNNV?
Figure 3E: according to this figure, only 0.4% of cells inoculated with wt are expressing B2 protein. I find this percentage very low, can you explain this.
Figure 3F: explain somewhere in the MS how the “mean fluorescence intensity of B2 positive cells” has been calculated.
Figure 3G: since results about B2/actin are shown in this figure, I guess that band intensity has been measured, am I right? if so, please indicate in "Material and Methods" the software used.
Figures 3H, 3I: there is no information about these experiments in “Material and Methods”.
Why the wild type has not been included in this experiment? In my opinion, all the experiments shown in Figure 3 should include this control, not only some of them.
Zebrafish experiments:
Line 314: at what time p.i was the viral load and the cytokine expression tested? Add this information in “Material and Methods”.
Line 323: M2 mutant infection caused a significant decrease in CRP transcription compared to wtRGNNV; however, Figure 4B shows that this virus causes the same transcription as rRGNNV. How is this possible if wtRGNNV and rRGNNV are genetically identical?
Information about immunogene and fluorescence image analyses in zebrafish should be added in “Material and Methods”.
Why have you analysed different genes in SNN-1 cells and zebrafish? Are those gene relevant for the control of the disease? Add this information in the MS
Some methodological details included in this section are misplaced, move them to “Material and Methods”.
Did zebrafish larvae develop symptoms and/or mortality? This information should be added in the MS.
Figure 4D: the dose of this experiment should be added in “Material and Methods”. I am not sure if I have understood this scheme: did you kill larvae for immunostaining at 8 dpf? this information is missed in “Material and Methods”.
The results of the viral immunostaining shown in Figure 4E are not described in the MS. Do you have information about B2 expression in zebrafish brain? immunostaining using anti-B2 antibodies could also be conducted.
Discussion
The “Discussion” section should be re-written, since results have not been really discussed. The current discussion is just a list of possible applications of the reverse genetics procedure, and results have not been confronted with previous publication (there is only 3 references in this section). The results shown in this MS have arisen some question that should be discussed, such as: which is the role of the different immunogene expression after the infection with mutated viruses? I have also noticed that the multiplication of all viruses is the same (in vitro and in vivo); however, they caused different immunogene response (in vitro and in vivo) and different level of brain damage in zebrafish. This is a very interesting result that should be further discussed (which are the implications of these differences? are those viruses less or more virulent?). By looking at Figure 4E, it seems clear that, somehow, this protein is responsible for brain damage, how is this related to the classical role assigned to B2 protein (inhibition of cellular RNA-interference mechanism)?
Lines 368-370: authors compare the reverse protocol they have developed with previous protocols. I can see the advantage of green fluorescence, but which is the advantage of using SNN-1 cells instead of E-11 cells?
Lines 379-380: there is a mistake in this phrase, since the kinetics of rRGNNV shown in Figure 2F is the same as the kinetics of mutant viruses.
Line 382: it is written “B2 mutant probably affected the expression of viral RdRp protein”. Authors can easily prove this hypothesis by analyzing samples from Figure 2F using specific primers for RNA1 segment.
Paragraph 389-405: this paragraph is not really a discussion, is a “future work” section. It should be considerably reduced and be part of the “conclusion” paragraph.
Author Response
The manuscript entitled “Construction of attenuated strains for red-spotted grouper nervous necrosis virus (RGNNV) via reverse genetic system”, describes the generation of two recombinant RGNNV viruses with point mutations within the B2 sequence. Authors have stablished the reverse genetics protocol to generate those viruses and have tested their infectivity both in vitro and in vivo. The manuscript is well written, although some minor mistakes regarding the use of the English language and important mistakes regarding organization have been detected. I suggest to read carefully the MS to correct English mistakes and to consider the following comments:
Thank you very much for your valuable comments and suggestion. We have made revision of our manuscript according to your suggestion and responded to all the comments in this letter. All changes have been tracked in our revised manuscript. Please see our point-by-point response in the following.
General comment
I suggest not to use the term “mRNA” or “transcription” when you refer to viral RNA quantification, RGNNV is a positive RNA virus, and, therefore, distinction between genomic RNA and mRNA (or replication and transcription) is not really possible.
Thank you for the suggestion! We have replaced the “mRNA” with “RNA” for viral RNA quantification.
Specific comments
Introduction
Line 45: this information is incomplete, since RNA2 also encodes B1 protein. B1 is also missed in line 229.
Thank you for the suggestion! We have added the description of B1 protein (“A subgenome of RNA1, named RNA3, located at the 3’-terminus of RNA1, encodes two small non-structural proteins B1 (111 aa) and B2 (75 aa)” in line 44-45 and “Of four proteins (RdRp, Cp, B1, B2) expressed by RGNNV, the non-structural protein B2 plays an important role in viral replication and cytopathy.” in line 278 of our revised manuscript).
Line 48: it is written that “commercially available vaccine against RGNNV is currently lacking”. To my knowledge, there are at least two commercial formalin-inactivated vaccines against RGNNV: ALPHA JECT micro® 1Noda (Paharmaq) and ICTHIOVAC® VNN (Hipra), both for sea bass vaccination.
Thank you! We have corrected the description as “Meanwhile, commercial vaccine against RGNNV is still desperately needed, and potential vaccine candidates against RGNNV for large-scale application in fishes are still at laboratory level” in our revised manuscript (line 59-62).
Line 59: authors claim that the reverse genetic system for RGNNV has not being established. This information is not accurate, recombinant RGNNV viruses have been obtained by reverse genetics by Moreno et al. (2019), and recombinant RGNNV/SJNNV reassortant viruses have also being retrieved by Souto et al. (2015, 2018). As a general comment, a more rigorous reference searching is required in the whole MS.
Thank you for the correction! We have corrected the claim and also added the references of reverse genetics for RGNNV as “Both RNA-based and DNA-based genetic systems are now available for several betanodavirus species including RGNNV [9, 33-37]. However, the reverse genetic system for attenuated strains of RGNNV has not been established, hampering pathogenic mechanism study and vaccine development for RGNNV. Thereby, it is of great necessity to develop reverse genetic system for the construction of attenuated strains of fish betanodavirus RGNNV.” in our revised manuscript (line 69-74).
In this work the transcription of some immunogenes has been analysed in vitro and in vivo; however, information about the relevance of those genes in controlling NNV infection does not appear in the “Introduction” section. This information should be added. The transcription analysis of those genes should be put in contest.
We have added this information in the introduction part of our revised manuscript (line 49-58).
Materials and methods
This section is specially confusing, and important information is missing.
“Antibodies and reagents” section (line 85): if anti-B2 antibodies have been already published, authors should add that reference; if not, details about B2 expression, rabbit immunization procedure, and antibody purification should be added.
We asked the aBIOTECH company (Jinan, China) to produce the B2 antibody. They expressed the full length B2 protein in E. coli and use it to immunized rabbit, finally obtained the anti-B2 polyclonal antibodies. The affinity and specificity of B2 antibody were tested by western blotting assay as shown in the following figure. The band for B2 was detected by anti-B2 antibody in wild-type RGNNV (wtRGNNV) infected SSN-1 cells but not in non-infected (Mock) SSN-1 cells. We have added the description on the B2 antibody production and confirmation in the method part (line 99-103) and the following figure as new supplemental Figure 1 in our revised manuscript.
“Construction of RNA1 and RNA2 expression plasmids” (line 92): I miss information about the sequencing strategy and primers used for RNA1 and RNA2 sequencing, as well as about the cloning procedure. It may be convenient to add this information as supplementary material.
Thank you for the suggestion! We asked Shanghai Sangon Biotech company to confirm the inserted sequence by Sanger sequencing. They used the upper primer of T7 promoter (T7p-F: ctcagatcttaatacgactcactatagg) and terminal primer of inserted fragment (RNA1-T7t-R: taccgaattcccgctgagcaataactagc; RNA2-T7t-R: taccgaattcccgctgagcaataac) to perform sequencing. The fragment of RNA1-HDVrz and RNA2-HDVrz were cloned into vector through double enzyme digestion of BglII and EcoR1 sites. We have added this information into methods part (line 116-118).
“Virus infection and titration” (line 121):
Please include information about the MOI used for cell inoculation, and the specific sampling times. This information is in “Results” and should appear in “Material and Methods” section.
As suggested, we have added the information in methods part of our revised manuscript (“For virus growth ability assay, SSN-1 cells were incubated with wtRGNNV, rRGNNV, rRGNNV-B2-M1 or rRGNNV-B2-M2 virus at an MOI of 5. Then Cells were collected at 0, 24, 48 and 72 h poi for viral Cp RNA detection by RT-qPCR with primers listed in Table 1. Three replicates for each treatment were analyzed.”) in line 147-150.
This section is named “virus infection and titration”; however, information about RT-qPCR, and western-blot is included. I suggest to either move that information to another section or to change the heading of the section. Nevertheless, information about the sampling time for RT-qPCR and blot analysis should be added.
We have changed the title of “virus infection and titration” to “virus infection, titration and growth ability assay”. Meanwhile, we moved the western-blot information with sampling time into the section of Western blotting assay. Since RT-qPCR is for virus titration, we remained the information in original section.
Line 124: it is said that viral titration has been performed at different times poi, but these results are not in the MS.
As suggested, we have added the information of poi time for viral titration in methods part of our revised manuscript (“Then Cells were collected at 0, 24, 48 and 72 hpi for viral Cp RNA detection by RT-qPCR with primers listed in Table 1”) in line 148-150.
I do not see the point of expressing nodavirus titre as PFU. I think it would be more adequate to express it as TCID50/ml, especially considering that nodavirus does not form plaques on cell monolayers. The relationship between PFU and TCID50 in line 127 is widely used for adenovirus, but it may not be correct to use it for nodavirus. Finally, using TCID50 would allow to compare your results with previous publications, which express nodavirus titres as TCID50.
Thank you for the suggestion! We totally agree with you. As suggested, we have corrected PUF into TCID50/mL for RGNNV titer in our revised manuscript.
Table 1: primers for endogenous gene amplification should be added.
As suggested, we have added the primers for endogenous gene β-actin in revised table 1.
Table 1 heading: it is written “primers for RT-PCR and RT-qPCR”; however, there is only one pair of primers for the amplification of RNA2 genomic segment. Were those primers used for both, PCR and qPCR, or there is a pair of primers missed in this table? Furthermore, there are two pair of primers for RT reaction (CP and RdRp), whereas only the pair of primers for CP amplification is included.
We used the pairs of RNA1 primers (RdRp-RT-F and RdRp-RT-R) and RNA2 primers (Cp-RT-F and Cp-RT-R) respectively for RT-PCR of RNA1 and RNA2 in Fig.3C. In Fig.4A, we used primer pair for RNA2 (Cp) to perform RT-qPCR.
“RNA extraction, virus RT-PCR and RT-qPCR assays” (line 129):
Line 130: it is said that RNA is extracted at 3 days poi and cDNA was amplified with primers in Table 1 (for viral genome amplification, line 141); however, I have not found results of viral qPCR from samples taken at that time. The only qPCR results are those displayed in Figure 2F (samples collected at 0, 24, 48 and 72 h).
In Figure 2F, samples were collected at 0, 24, 48 and 72 hpi. In Figure 3H and I, samples were collected at 3 dpi. In Figure 4A-4C, zebrafish samples were collected at 30 hpi. All indicated time points have been described in our manuscript. We have rewritten the description of RNA extraction (“Total RNA was isolated from RGNNV-infected SSN-1 cells or zebrafish at indicated time point using Trizol reagent (Invitrogen, 15596026) according to the manufacturer’s instructions. For virus growth ability assay, total RNA was isolated from RGNNV-infected SSN-1 cells at 0, 24, 48 and 72 hpi. For RNA1, RNA2 and cytotoxic protein (Endo-G and Mx1) analysis, total RNA was isolated from RGNNV-infected SSN-1 cells at 3 days post-infection (dpi). And for in vivo virulence analysis, zebrafish at 30 hpi were sampled for RNA extraction.”) in line 153-159 of revised manuscript.
Line 141: it is written that the internal control of amplification is “grouper beta-actin”. This may be a mistake, since SSN-1 cells do not derive from that fish species.
Thanks for the valuable comments.18S and zebrafish β-actin were used as internal control of the amplifications for SSN-1 cells and zebrafish, respectively. We have added 18S internal control in the revised manuscript (in lines 170).
“Western blotting assay” (line 145):
According to line 146, western blotting was conducted on B7GG and SSN-1 cells; however, this information does not agree with that in “virus infection and titration” section, where it is said that western blotting is perform on SSN-1 cells, and B7GG cells are not mentioned.
SSN-1 cells were used for virus titration and western blotting of RGNNV, while B7GG cells were used for testing B2 expression of constructed RGNNV plasmids. As mentioned above, we have moved the western-blotting information with sampling time into the section of Western blotting assay in our revised manuscript (“B7GG cells at 48 h post of transfection and SSN-1 cells at 3 dpi were washed with PBS and then lysed with RIPA lysis buffer”) line 180.
According to lines 147-148, SNN-1 cells were transfected; this information does not agree with that in line 247, where it is said: “for positive control, SNN-1 cells were infected with RGNNV directly”.
For western blotting, B7GG cells were transfected with plasmid, and SSN-1 cells were infected with virus.
At what time post-transfection or post-infection were the cells collected for western blotting?
B7GG cells at 48 h post of transfection and SSN-1 cells at 3 days poi were collected for Western blotting. We have added the time point in line 180 of revised manuscript.
“Zebrafish experiment” (line 178):
In line 181 it is said “during viral injection, embryos……” and in line 183 “virus infection of zebrafish larvae….”. did you challenge embryos in addition to larvae?
Sorry for this descriptive error, we have corrected embryo into larvae in our revised manuscript.
Add information about the challenge: viral dose (which is in “Results” section), number of larvae, number of replicas, times of sampling, strain of zebrafish…..
As suggested, we have added the information including viral dose, number of larvae, number of replicas, and strain of zebrafish in our revised manuscript (line 215-220).
Nothing is said in this section about the challenge conducted with the transgenic zebrafish strain or about the immunogene analyses, add this information in this section.
As suggested, we have added the detailed information about the challenge with transgenic zebrafish in our revised manuscript (line 221-230).
Results
Figure 2D: how was the B2 protein calculated in this graph? I have not found this information in the MS.
For quantification analysis, bands for B2 and β-actin were quantified using the threshold function of Image J. Three independent experiments for B2 western blotting were analyzed for quantification. Statistical significance was analyzed using the Student's t‐test (*P < 0.05; **P < 0.01; n.s.=not significant). We have added this information in methods part of our revised manuscript (line 193-196).
Figure 2E: in line 256 and figure legend is said “virus titre by TCID50”. More explanation is required: what has been this titre obtained from? are they samples obtained at a specific time post-infection? Why there is no SD in this graph?
We detected virus titer by TCID50 and calculated as PFU with formula: PFUs=0.7xTCID50 in our first version of manuscript. Since nodavirus does not form plaques on cell monolayers, we directly used TCID50/mL for virus titer of wtRGNNV, rRGNNV, rRGNNV-B2-M1 and rRGNNV-B2-M2 in our revised manuscript. Thus, in revised Figure 2E (shown in the following), there is SD in the graph.
For virus titration, SSN-1 cells were incubated with wtRGNNV, rRGNNV, rRGNNV-B2-M1 or rRGNNV-B2-M2 virus. At 72 h post-infection (hpi), both the supernatants and cells were collected for viral titration by TCID50.
Figure 2F: samples were taken at different times, and in line 130 it is said at 3 days poi.
Sorry for incomplete information on sampling time in methods part. As answered above, we have rewritten the description of sampling time for RNA extraction line 155-156 of revised manuscript.
Standard deviation should appear in this graph. Experimental replicas are required for statistical analyses.
Since the error bars were shorter than the size of symbols in Figure 2F, it is hard to recognized error bars. In our revised Figure 2F, we removed all symbols so that the error bars can be recognized now.
Line 262: it is written that mutant recombinant viruses grow “almost equally” as wtRGNNV. In order to stablish if there are differences in viral multiplication, a statistical test should be used.
Yes, we have did Student t-test for statistical analysis and added the significance in revised Figure 2F.
I do not understand the difference between the copy number of wtRGNNV and rRGNNV at 24 h p.i. Both viruses are supposed to be genetically identical, and, therefore, they should grow with the same kinetics in SNN-1 cells. Please, explain this result.
Yes, we also found rRGNNV grows slower than wtRGNNV when we replicated the virus each time. We have sequenced the rRGNNV and didn’t find any mutation. There might me some alteration (like RNA epigenetic modification) except sequence in rRGNNV as compared with wtRGNNV, which need to be unraveled. But we still don’t know the exact reason.
Figure 3A: did the authors need a blind passage with any of the recombinant viruses?
Yes. When we rescued rRGNNV, rRGNNV-B2-M1 and rRGNNV-B2-M2 viruses, we performed three blind passages in SSN-1 cells using cell lysates from transfected B7GG cells.
Figure 3B: please, explain what is the “cytopathic number” and how it has been calculated
To measure the cytopathic effect of rRGNNV, rRGNNV-B2-M1 and rRGNNV-B2-M2, we infected SSN-1 cells with the three constructed viruses with non-infected SSN-1 cells as control. For each virus infection, three repeats were obtained. Then we randomly selected three visual fields to capture the images for each culture well under 40X microscope. We counted the cell number of vacuolar cytopathic lesions in each image for final statistical analysis.
Why the wtRGNNV group is included in Figure 3A and disappears in Figure 3B?
Because wtRGNNV caused quite severe cytopathic effect than recombinant RGNNV at 3 dpi, it is hard to accurately quantify the exact cytopathic number. Thus, to be stricter, we didn’t analyze wtRGNNV.
Figure 3C: primers used to amplify RNA2 are missed in Table 1.
The primer pair for RNA2 amplification is Cp-RT-F and Cp-RT-R since RNA2 encodes capsid protein (Cp) sequence. So, the primers have been listed in Table 1.
Based on this picture, authors claim that the expression of RNA1 and RNA2 was not affected by mutations (line 295). Although this seems to be correct, in my opinion this study should have been conducted by qPCR or, alternatively, the analysis of the intensity of the bands should have been performed. If that analysis has been conducted, this information should appear in “Material and Methods”. Why have you chosen conventional PCR instead of q-PCR in this assay?
We agree with you. Figure 3C mainly aimed to confirm the infection of recombinant RGNNV (rRGNNV, rRGNNV-B2-M1 and rRGNNV-B2-M2) in SSN-1 cells to exclude the possibility of non-infection induced less CPE in rRGNNV-B2-M1 and rRGNNV-B2-M2. So, we only did RT-PCR. Meanwhile, due to loss of internal control, we can’t do analysis of band intensity. However, according to RT-qPCR results in Figure 2F, no significant difference on the copy number of RNA2 for rRGNNV-B2-M1, rRGNNV-B2-M2 and rRGNNV at 72 hpi (3 dpi) were observed. To be stricter, we have changed the claim to “The expression of RGNNV RNA1 and RNA2 were detected in SSN-1 cells infected with rRGNNV-B2-M1, rRGNNV-B2-M2, and wtRGNNV detected by RT-PCR (Figure. 3C).” in line 339-342 of our revised manuscript.
Were those samples collected at 3 dpi? If so, there is a contradiction between the results of RNA2 transcription shown in Figure 3C and Figure 2F. According to Figure 2F, at 3 d p.i. the copy number of RNA2 is similar for all viruses, including the wild-type, whereas the band for the wild type virus is clearly more intense in Figure 3C. Can you explain this?
Yes, the cell samples were collected at 3 dpi. In Figure 2F, the copy number of RNA2 is lower in recombinant RGNNV (rRGNNV, rRGNNV-B2-M1, rRGNNV-B2-M2) than in wtRGNNV at all times we tested (24 hpi, 48 hpi, and 72 hpi). As we explained above, we can’t accurately quantify band intensity for this RT-PCR results in Figure 3C. However, we speculate that the RNA1 and RNA2 should be higher in wtRGNNV than in recombinant RGNNV due to higher growth ability of wtRGNNV.
Figure 3D: in line 283 it is said that immunofluorescence has been performed at 48 h pi, is that correct? if so, please correct the information in “Material and Method”, where it is said that this technique was applied 24 h pi (line 167).
According to our description on the time point in methods, the immunofluorescence was performed at 36 hpi. We have corrected it.
I think that the expression of B2 should be the same in cells inoculated with wtRGNNV and rRGNNV, why is lower for rRGNNV?
According to significantly lower copy number of rRGNNV than wtRGNNV in Figure 2F, rRGNNV grew slower than wtRGNNV, which should be main reason for lower B2 expression in rRGNNV as compared with wtRGNNV.
Figure 3E: according to this figure, only 0.4% of cells inoculated with wt are expressing B2 protein. I find this percentage very low, can you explain this.
The ordinates value is ratio value for B2 positive cells, we have corrected the ratio value to percent value.
Figure 3F: explain somewhere in the MS how the “mean fluorescence intensity of B2 positive cells” has been calculated.
The description on mean fluorescence intensity analysis can be found in the methods part of revised manuscript (line 208-211).
Figure 3G: since results about B2/actin are shown in this figure, I guess that band intensity has been measured, am I right? if so, please indicate in "Material and Methods" the software used.
As suggested, we have added the description on band intensity quantification in the methods part of revised manuscript (line 193-196).
Figures 3H, 3I: there is no information about these experiments in “Material and Methods”.
We have added the information in the methods part of revised manuscript (line 156-158).
Why the wild type has not been included in this experiment? In my opinion, all the experiments shown in Figure 3 should include this control, not only some of them.
Thank you for the suggestion. We have added the wtRGNNV data in revised Figure 3H-3I.
Zebrafish experiments:
Line 314: at what time p.i was the viral load and the cytokine expression tested? Add this information in “Material and Methods”.
As suggested, we have added this information in methods part in our revised manuscript (line 215-220).
Line 323: M2 mutant infection caused a significant decrease in CRP transcription compared to wtRGNNV; however, Figure 4B shows that this virus causes the same transcription as rRGNNV. How is this possible if wtRGNNV and rRGNNV are genetically identical?
Although wtRGNNV and rRGNNV are genetically identical, the growth ability, cytopathic effect, B2 level of rRGNNV are all lower than that of wtRGNNV in our results. Based on statistical analysis, there is no difference between rRGNNV-B2-M2 versus rRGNNV and rRGNNV versus wtRGNNV, but significant difference between rRGNNV-B2-M2 and wtRGNNV for CRP transcription level. Although no difference between rRGNNV versus wtRGNNV, we can see a tendency of decrease for rRGNNV as compared with wtRGNNV. The variation bar is a little big for both rRGNNV and wtRGNNV in Figure 4B-4C, which is likely caused the difference between wtRGNNV and rRGNNV is not significant.
Information about immunogene and fluorescence image analyses in zebrafish should be added in “Material and Methods”.
As suggested, we have added this information in methods part in our revised manuscript (line 227-230)
Why have you analysed different genes in SNN-1 cells and zebrafish? Are those gene relevant for the control of the disease? Add this information in the MS
Mx1 is antiviral pathway related protein and Endo-G is cytotoxic protein. For cytopathic effect induced by RGNNV in SSN-1 cells, we analyzed Mx1 and Endo-G. In zebrafish, we selected two important inflammation factors to analyze the virulence of RGNNV in vivo. We have added more background on these genes in NNV pathogenicity in the Introduction part of revised manuscript (line 49-58).
Some methodological details included in this section are misplaced, move them to “Material and Methods”.
As suggested, we have moved them into methods part of our revised manuscript.
Did zebrafish larvae develop symptoms and/or mortality? This information should be added in the MS.
We didn’t observe obvious symptoms and mortality in zebrafish at 30 hpi by immersion infection and at 4 dpi by micro-injection infection. We have added this information in methods part (line 231-233 of revised manuscript).
Figure 4D: the dose of this experiment should be added in “Material and Methods”. I am not sure if I have understood this scheme: did you kill larvae for immunostaining at 8 dpf? this information is missed in “Material and Methods”.
As suggested, we have added this information in methods part in our revised manuscript (line 221-230).
The results of the viral immunostaining shown in Figure 4E are not described in the MS. Do you have information about B2 expression in zebrafish brain? immunostaining using anti-B2 antibodies could also be conducted.
We didn’t do B2 immunostaining but did Cp immunostaining in transgenic zebrafish brain.
Discussion
The “Discussion” section should be re-written, since results have not been really discussed. The current discussion is just a list of possible applications of the reverse genetics procedure, and results have not been confronted with previous publication (there is only 3 references in this section). The results shown in this MS have arisen some question that should be discussed, such as: which is the role of the different immunogene expression after the infection with mutated viruses? I have also noticed that the multiplication of all viruses is the same (in vitro and in vivo); however, they caused different immunogene response (in vitro and in vivo) and different level of brain damage in zebrafish. This is a very interesting result that should be further discussed (which are the implications of these differences? are those viruses less or more virulent?). By looking at Figure 4E, it seems clear that, somehow, this protein is responsible for brain damage, how is this related to the classical role assigned to B2 protein (inhibition of cellular RNA-interference mechanism)?
Thank you for the suggestion! We have added more discussion about our results in revised manuscript.
Lines 368-370: authors compare the reverse protocol they have developed with previous protocols. I can see the advantage of green fluorescence, but which is the advantage of using SNN-1 cells instead of E-11 cells?
E-11 cells is a clone of SSN-1 cells and is persistently infected with a C-type retrovirus (SnRV). We have SSN-1 cells in our lab and SSN-1 is good for RGNNV replication, so we used SSN-1 cells for our reverse protocol. We just want to point out the difference between our reverse protocol and others, although EGFP is an advantage. To avoid misleading, we have deleted the description on cells.
Lines 379-380: there is a mistake in this phrase, since the kinetics of rRGNNV shown in Figure 2F is the same as the kinetics of mutant viruses.
Sorry for the inaccurate description. To better present statistical difference, we added the P value for rRGNNV-B2-M1 vs rRGNNV and rRGNNV-B2-M2 vs rRGNNV in revised Figure 2F. As shown in Figure 2F, the copy number of rRGNNV-B2-M1 at 48 hpi and rRGNNV-B2-M2 at 24 hpi is significantly lower than rRGNNV. We have re-written the phrase (line 446-447) and also corresponding description in the results part (line 325-329) to avoid misleading in revised manuscript.
Line 382: it is written “B2 mutant probably affected the expression of viral RdRp protein”. Authors can easily prove this hypothesis by analyzing samples from Figure 2F using specific primers for RNA1 segment.
It’s a good suggestion. We didn’t do the RT-qPCR for RdRp when we measured Cp RNA level for analyzing viral growth ability. In future, we can test it.
Paragraph 389-405: this paragraph is not really a discussion, is a “future work” section. It should be considerably reduced and be part of the “conclusion” paragraph.
Thank you for your suggestion! We have considerably reduced this paragraph but remained it in discussion. Meanwhile, we also added a short sentence on the potential of attenuated rRGNNV for developing neural tracer in conclusion part (“It is also promising to developing neural tracer based on attenuated rRGNNV for neuroscience study.” In line 497-498).

Reviewer 3 Report
The article "Construction of attenuated strains for red-spotted grouper nervous necrosis virus (RGNNV) via reverse genetic system" describes in-vitro and in-vivo research on red-spotted grouper nervous necrosis virus. The authors present their interesting experiments on the construction of attenuated RGNNV strains and their virulence in zebrafish. It is a valuable manuscript, which should be published after minor corrections.
In the introduction, the authors describe the virus and its molecular characteristic.
In line 35 - “betanodavirus” should be in italic.
In line 64 - shortcuts “B7GG” and “SSN-1” should be explained.
In lines 63-68 authors should describe the main goal of the study - I recommend improving this part.
Materials and methods
Line 86 - E. coli - in italic
Lines 87 and 89 - explain shortcuts
Line 98 - the sequences should be marked in Figure S1 and a description in Supplementary Materials should be added
Line 123 - I recommend using rather a day post-infection (dpi) or hour post-infection (hpi) than a day post of infection (poi) or h poi
Line 176 - “and” not “And”
Results
Line 197 - Figure S2, not S1 in this place authors should add that plasmid maps are presented in Supplementary Figure S2.
Figure 1F - the title of these figures should be added because these photos are almost identical and description under the Figure 1 is not enough
All Figures, Schemes, and Tables should have a short explanatory title and caption
Figure 3 - All Figures, should be inserted into the main text close to their first citation and must be numbered following their number of appearance
All figures and tables should be cited in the main text as Figure 1, Table 1, etc., not Fig.1, etc.
Line 330 - the cytopathic effect is observed rather in-vitro than in-vivo. If you observed any phenotypic effect in zebrafish larvae please describe it.
Figure 4A - in the caption please explain the “Cp” shortcut, which is on the figure, and also the “ns” shortcut
Discussion
Lines 406-412 - I recommend marking as part 6. Conclusions
Authors should discuss the results and how they can be interpreted from the perspective of previous studies and the working hypotheses. In my opinion, should be more references to other publications.
Supplementary Materials
Add Figure S1 - the sequences of RNA1 and RNA2
Change Figure S1 on Figure S2 - plasmid maps
Author Response
The article "Construction of attenuated strains for red-spotted grouper nervous necrosis virus (RGNNV) via reverse genetic system" describes in-vitro and in-vivo research on red-spotted grouper nervous necrosis virus. The authors present their interesting experiments on the construction of attenuated RGNNV strains and their virulence in zebrafish. It is a valuable manuscript, which should be published after minor corrections.
Thank you very much for your comments and suggestion. We have made revision of our manuscript according to your suggestion and responded to all the comments in this letter. All changes have been tracked in our revised manuscript. Please see our point-by-point response in the following.
In the introduction, the authors describe the virus and its molecular characteristic.
In line 35 - “betanodavirus” should be in italic.
As suggested, we have corrected it.
In line 64 - shortcuts “B7GG” and “SSN-1” should be explained.
In methods part, we explained two cell types. “B7GG cells, which are originated from BHK-21 cells and express T7 RNA polymerase, rabies virus G and histone-tagged GFP [38]” in line 84-85 and “Striped snakehead fish (SSN-1) cells” in line 90-91 of our revised manuscript.
In lines 63-68 authors should describe the main goal of the study - I recommend improving this part.
As suggested, we have described the main goal of our study in revised manuscript (line 75-76).
Materials and methods
Line 86 - E. coli - in italic
As suggested, we have corrected it.
Lines 87 and 89 - explain shortcuts
As suggested, we have explained the shortcuts in revised manuscript (“Horseradish Peroxidase (HRP)” in line 104 and “4’,6-diamidino-2-phenylindole (DAPI)” in line 106).
Line 98 - the sequences should be marked in Figure S1 and a description in Supplementary Materials should be added
As suggested, we have added sequences for RNA1 and RNA2 as new Figure S2 and a description for it in supplementary materials of revised manuscript.
Line 123 - I recommend using rather a day post-infection (dpi) or hour post-infection (hpi) than a day post of infection (poi) or h poi
As suggested, we have replaced day post of infection (poi) or h poi by day post-infection (dpi) or hour post-infection (hpi).
Line 176 - “and” not “And”
We have deleted “and” in the revised manuscript.
Results
Line 197 - Figure S2, not S1 in this place authors should add that plasmid maps are presented in Supplementary Figure S2.
As suggested, we have added the sequence of RNA1 and RNA2 as supplementary Figure 2 in our revised manuscript.
Figure 1F - the title of these figures should be added because these photos are almost identical and description under the Figure 1 is not enough
As suggested, we have added title in revised Figure 1F and added description for Figure 1.
All Figures, Schemes, and Tables should have a short explanatory title and caption
Yes, all Figures and Table in our manuscript have title and captions.
Figure 3 - All Figures, should be inserted into the main text close to their first citation and must be numbered following their number of appearance
Yes, we have adjusted all Figures to their main text close to their first citation.
All figures and tables should be cited in the main text as Figure 1, Table 1, etc., not Fig.1, etc.
As suggested, all Figures and Table have been cited in the main text with full name but not abbreviation in revised manuscript.
Line 330 - the cytopathic effect is observed rather in-vitro than in-vivo. If you observed any phenotypic effect in zebrafish larvae please describe it.
For all zebrafish infected with wtRGNNV and recombinant RGNNV at 30 hpi by immersion infection or at 4 dpi by micro-injection infection, no obviously abnormal swimming behavior and mortality were observed in these experiments.
Figure 4A - in the caption please explain the “Cp” shortcut, which is on the figure, and also the “ns” shortcut
As suggested, we have added this information in caption for Figure 4 (in line 410 of revised manuscript).
Discussion
Lines 406-412 - I recommend marking as part 6. Conclusions
As suggested, we have made this paragraph as Conclusion part.
Authors should discuss the results and how they can be interpreted from the perspective of previous studies and the working hypotheses. In my opinion, should be more references to other publications.
As suggested, we have added more discussion on our results in discussion part of revised manuscript.
Supplementary Materials
Add Figure S1 - the sequences of RNA1 and RNA2
As suggested, we have added Figure S2 on the sequences of RNA1 and RNA2 in our revised manuscript.
Change Figure S1 on Figure S2 - plasmid maps
As suggested, we have changed Figure S1 on the plasmid maps to Figure S3.

Round 2
Reviewer 2 Report
The quality of the manuscript entitled “Construction of attenuated strains for red-spotted grouper nervous necrosis virus (RGNNV) via reverse genetic system” has considerably increased. I suggest to make further minor changes:
General comment
Authors have introduced the expression “growth ability” in several parts of the MS. I would replace this expression by “viral replication” or kinetics of “viral replication”.
There are minor mistakes throughout the MS, such as grammatical mistakes, scientific names non-written in italics, words written in capital letters with no reason……These mistakes should be solved.
Specific comments
Introduction
Line 71: authors claim that “the reverse genetic system for attenuated strains of RGNNV has not been established”. This information is not accurate, recombinant RGNNV viruses obtained by Moreno et al. (2019) were attenuated, causing significantly less mortality that the wild-type RGNNV virus.
Materials and methods
“RNA extraction, virus RT-PCR and RT-qPCR assays” (line 152):
This section should contain only information on RNA extraction, RT-PCR and RT-qPCR. It does not make sense to show information on zebrafish samples or immunogene analyses without having described those experiments yet. This information should be moved to somewhere else.
Information in lines 155-156 can be deleted, since in has already been described in lines147-150.
“Zebrafish experiment” (line 212):
In line 216, Danio rerio should be italics.
The last paragraph of this section (about the lack of symptoms or mortality) should be moved to “results” section.
Results
Figure 2C-D (figure legend): it is said that western blot was performed at 36 h pi, whereas in line 180 it is written that is at 48 h pi.
Figure 3A: did the authors need a blind passage with any of the recombinant viruses?
Authors answered to this question in their letter. I think it would be convenient to add the information they gave me in the MS. This may be a useful information for readers.
Figure 3B: please, explain what is the “cytopathic number” and how it has been calculated. Authors answered to this question in their letter. I think it would be convenient to add the information they gave me in the MS. This may be a useful information for readers.
Figure 3D (figure legend): in line 363 it is said that immunofluorescence has been performed at 48 h pi, when, according to the author letter, it was changed to 36 h pi.
Figures 3H, 3I: there is no information about these experiments in “Material and Methods”. Authors have added this information, although, as I have mentioned before, it is not in the correct place.
New Figures 2E-2F: the nomenclature of the recombinant virus has been changed in Figure 2E.
The meaning of “###” does not appear in the figure legend. I do not understand the symbols appearing next to rRGNNV-B2-M1 (*) and rRGNNV-B2-M2 (#).
The statistical analysis should include wtRGNNV.
Zebrafish experiments:
Lines 378, 409: authors refer to “viral titre” in zebrafish, when they did not make any titration. Instead, they have quantified viral genome. This mistake should be solved
Author Response
The quality of the manuscript entitled “Construction of attenuated strains for red-spotted grouper nervous necrosis virus (RGNNV) via reverse genetic system” has considerably increased. I suggest to make further minor changes:
Thank you very much for your valuable comments and suggestion. We have made further revision according to your suggestion. All changes have been tracked in our new revised manuscript. Please see our point-by-point response in the following.
General comment
Authors have introduced the expression “growth ability” in several parts of the MS. I would replace this expression by “viral replication” or kinetics of “viral replication”.
As suggested, we have changed “growth ability” into “kinetics of viral replication”.
There are minor mistakes throughout the MS, such as grammatical mistakes, scientific names non-written in italics, words written in capital letters with no reason……These mistakes should be solved.
Thank you for the suggestion! We have carefully corrected these typing errors.
Specific comments
Introduction
Line 71: authors claim that “the reverse genetic system for attenuated strains of RGNNV has not been established”. This information is not accurate, recombinant RGNNV viruses obtained by Moreno et al. (2019) were attenuated, causing significantly less mortality that the wild-type RGNNV virus.
Thank you for the suggestion! We have re-written the sentence into “The recombinant RGNNV harboring site-specific mutations in the capsid protein sequence caused a significant decrease in virulence [35]. Due to the important role of B2 protein [9-12], constructing RGNNV without B2 expression will be another potential attenuated strain for vaccine development and pathogenic mechanism study of RGNNV. However, the reverse genetic system for attenuated strains of recombinant RGNNV lacking B2 protein has not been successfully established.” in line 70-75 of revised manuscript.
Materials and methods
“RNA extraction, virus RT-PCR and RT-qPCR assays” (line 152):
This section should contain only information on RNA extraction, RT-PCR and RT-qPCR. It does not make sense to show information on zebrafish samples or immunogene analyses without having described those experiments yet. This information should be moved to somewhere else.
As suggested, we have moved RT-qPCR assay for zebrafish samples to 2.11 Zebrafish experiment as “Then total RNA was isolated from zebrafish larvae at 30 hpi and followed by RT-qPCR with primers for Cp, CRP, and TNF-α listed in Table 1. Zebrafish β-actin was used as internal control.” in line 231-233 of revised manuscript. Meanwhile, we also moved RT-qPCR assay for Endo-G and Mx1 to the section “2.10. Cytopathic effect measurement and expression analysis on Endo-G and Mx1” in line 220-223.
Information in lines 155-156 can be deleted, since in has already been described in lines147-150.
The description in line 147-150 indicated the RNA extraction, while sentences for 155-156 described the further reverse transcription for RNA to cDNA. So, they are different. We kept the description.
“Zebrafish experiment” (line 212):
In line 216, Danio rerio should be italics.
We have change it into italics style (line 228 of revised manuscript).
The last paragraph of this section (about the lack of symptoms or mortality) should be moved to “results” section.
As suggested, we have moved this section into results part (line 384-385 and line 405-406 of revised manuscript).
Results
Figure 2C-D (figure legend): it is said that western blot was performed at 36 h pi, whereas in line 180 it is written that is at 48 h pi.
Thank you! We have corrected the error by change 48 hpi into 36 hpi.
Figure 3A: did the authors need a blind passage with any of the recombinant viruses? Authors answered to this question in their letter. I think it would be convenient to add the information they gave me in the MS. This may be a useful information for readers.
As suggested, we have added this description into Figure 3A and methods part (“The supernatant was subsequently incubated with fresh SSN-1 cells at 27 °C for three blind passages (3 days/passage)” in line 140-142 of revised manuscript).
Figure 3B: please, explain what is the “cytopathic number” and how it has been calculated. Authors answered to this question in their letter. I think it would be convenient to add the information they gave me in the MS. This may be a useful information for readers.
As suggested, we have added this description into methods part as “2.10. Cytopathic effect measurement and expression analysis on Endo-G and Mx1” (line 212-219 of revised manuscript).
Figure 3D (figure legend): in line 363 it is said that immunofluorescence has been performed at 48 h pi, when, according to the author letter, it was changed to 36 h pi.
We have corrected the error by change 48 hpi into 36 hpi.
Figures 3H, 3I: there is no information about these experiments in “Material and Methods”. Authors have added this information, although, as I have mentioned before, it is not in the correct place.
As suggested above, we have moved the information into the section for “2.10. Cytopathic effect measurement and expression analysis on Endo-G and Mx1” in methods part (line 220-223 of revised manuscript).
New Figures 2E-2F: the nomenclature of the recombinant virus has been changed in Figure 2E.
As suggested, we have corrected the error.
The meaning of “###” does not appear in the figure legend. I do not understand the symbols appearing next to rRGNNV-B2-M1 (*) and rRGNNV-B2-M2 (#).
We have added the meaning of “###” in the figure legend. Since we have added the notes on significance in the Figure 2F, we deleted the symbols next to rRGNNV-B2-M1 (*) and rRGNNV-B2-M2 (#) to avoid confusing in revised Figure 2.
The statistical analysis should include wtRGNNV.
As suggested, we have added the statistical analysis for rRGNNV, rRGNNV-B2-M1, or rRGNNV-B2-M2 vs wt RGNNV in results for Figure 2F (line 336-337 of revised manuscript).
Zebrafish experiments:
Lines 378, 409: authors refer to “viral titre” in zebrafish, when they did not make any titration. Instead, they have quantified viral genome. This mistake should be solved
We have corrected “viral titer” into “viral load” in legend for Figure 4A.
